# BLIPs: Bayesian Learned Interatomic Potentials

**Dario Coscia** [1 2]  **Pim de Haan** [3]  **Max Welling** [2 3]

## Abstract

Machine Learning Interatomic Potentials (MLIPs) are becoming a central tool in simulation-based chemistry. However, like most deep learning models, MLIPs struggle to make accurate predictions on out-of-distribution data or when trained in a data-scarce regime, both common scenarios in simulation-based chemistry. Moreover, MLIPs do not provide uncertainty estimates by construction, which are fundamental to guide active learning pipelines and to ensure the accuracy of simulation results compared to quantum calculations. To address this shortcoming, we propose BLIPs: Bayesian Learned Interatomic Potentials. BLIP is a scalable, architecture-agnostic variational Bayesian framework for training or fine-tuning MLIPs, built on an adaptive version of Variational Dropout. BLIP delivers well-calibrated uncertainty estimates and minimal computational overhead for energy and forces prediction at inference time, while integrating seamlessly with (equivariant) message-passing architectures. Empirical results on simulation-based computational chemistry tasks demonstrate improved predictive accuracy with respect to standard MLIPs, and trustworthy uncertainty estimates, especially in data-scarse or heavy out-of-distribution regimes. Moreover, fine-tuning pretrained MLIPs with BLIP yields consistent performance gains and calibrated uncertainties.

## 1. Introduction

Deep learning is transforming computational chemistry and materials science, enabling advances in molecular design (Özçelik et al., 2024; Eijkelboom et al., 2024; Zeni et al., 2023; Merchant et al., 2023), simulation-based modelling (Batatia et al., 2022; Wood et al., 2025; Neumann et al., 2024), and particle-based systems (Kipf et al., 2018; Alkin et al., 2024; Zhdanov et al., 2025). At the forefront of this progress are Message Passing Neural Networks (MPNNs) (Scarselli et al., 2008; Kipf and Welling, 2016; Gilmer et al., 2017), which have emerged as core architectures for learning in atomistic and molecular domains. Their ability to natively incorporate essential physical symmetries, such as translational, rotational, and permutation invariance or equivariance (Cohen and Welling, 2016; Weiler et al., 2023; Bronstein et al., 2017; Cohen et al., 2019), makes them particularly well-suited for modelling the complex interactions that define atomic-scale systems. Consequently, MPNNs are now widely adopted in the development of Machine Learning Interatomic Potentials (MLIPs), where they serve as efficient and accurate surrogates for quantum and classical simulations. These models enable large-scale atomistic simulations with near ab initio accuracy, dramatically accelerating research in materials discovery, catalysis, and molecular dynamics (Wood et al., 2025; Neumann et al., 2024; Batatia et al., 2022). However, like most deep learning models, MLIPs are vulnerable to failure, particularly when faced with out-of-distribution data. This issue is further worsened in scientific domains, where generating labelled data often requires expensive and time-consuming simulations (Papamarkou et al., 2024). Indeed, for simulation-based chemistry, training data is usually obtained from Density Functional Theory (DFT) calculations, which scale cubically with system size, making it impractical to generate data for larger molecules or materials, causing inevitable distributional shifts at test time (Özçelik et al., 2025; Tan et al., 2023). These challenges make principled Uncertainty Quantification (UQ) essential for enabling error-aware simulations or guiding active learning pipelines. Among existing approaches, Deep Ensembles (Lakshminarayanan et al., 2017) are commonly used for UQ in MLIPs and have shown strong empirical performance compared to other UQ methods (Tan et al., 2023). However, they require training and storing multiple models, making them computationally and memory-intensive, especially in large-scale or high-throughput scientific workflows. Simpler UQ methods, such as MC Dropout (Gal and Ghahramani, 2016) or mixture models, are less memory intensive but yield poorer uncertainty estimates and often worsen predictive accuracy.

---

[1]mathLab, SISSA, Italy [2]AMLab, University of Amsterdam, The Netherlands [3]CuspAI. Correspondence to: Dario Coscia <dario.coscia@sissa.it>.

*Proceedings of the 43rd International Conference on Machine Learning*, Seoul, South Korea. PMLR 306, 2026. Copyright 2026 by the author(s).

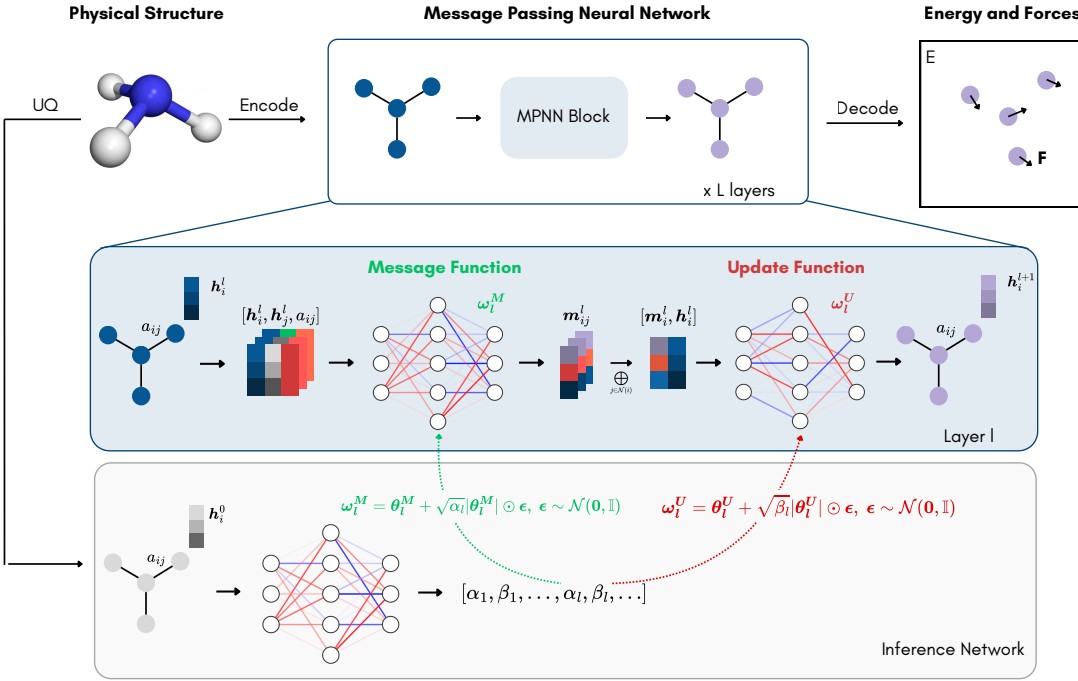

*Figure 1.* Bayesian Learned Interatomic Potentials (BLIP) for prediction and uncertainty Quantification. An atomic structure is encoded as a graph, with initial node features $\boldsymbol{h}^0$ and edge features $a_{ij}$, and processed by a standard Message Passing Neural Network. BLIP introduces stochasticity into the machine learned interatomic potential by injecting zero-mean Gaussian noise into the message and update functions. Specifically, the deterministic weights $\boldsymbol{\theta}_l^M$ and $\boldsymbol{\theta}_l^U$ are perturbed using perturbation scales $\alpha_l$ and $\beta_l$ (obtained through an inference network), respectively, yielding stochastic weights $\boldsymbol{\omega}_l^M$ and $\boldsymbol{\omega}_l^U$. The main weights $\boldsymbol{\theta}_l^M$ and $\boldsymbol{\theta}_l^U$, and the inference network weights are trained jointly using variational inference. This transforms MLIP into BLIP, a probabilistic model, enabling principled uncertainty quantification and improved predictive accuracy.

To overcome these limitations, we introduce *Bayesian Learned Interatomic Potentials (BLIPs)*, a general variational Bayesian method that is scalable, accurate, and provides principled, well-calibrated uncertainty estimates for MLIPs (see Figure 1). BLIP can be applied both during training and during fine-tuning of pretrained machine-learning interatomic potentials. It builds on an adaptive version of Variational Dropout (Kingma et al., 2015), where the MPNNs' weights are stochastically perturbed depending on the network's input data. BLIP does not depend on the chosen MPNN architecture and seamlessly integrates with a vast family of invariant and equivariant models. Importantly, our approach incurs minimal computational overhead relative to deterministic models during both training and predictive inference. We evaluate our approach on various simulation-based chemistry tasks, focusing on small and large-scale systems. We show that training with BLIP not only enhances predictive performance but also produces meaningful uncertainty estimates. Finally, we show that fine-tuning a single pretrained MLIP with BLIP consistently improves predictive performance over single or ensemble finetuning, while simultaneously providing good uncertainty estimates.

## 2. Background and Related Work

In this section, we present the relevant background on message passing neural networks, machine learning force fields, and uncertainty quantification, which will later support the formulation of our proposed method.

### 2.1. Message Passing Neural Networks

Message Passing Neural Networks (Scarselli et al., 2008; Kipf and Welling, 2016; Gilmer et al., 2017) are a class of (permutation equivariant) neural networks that process graph-structured data. Given a Graph $\mathcal{G} = (\mathcal{V}, \mathcal{E})$ with nodes $v_i \in \mathcal{V}$, and edges $e_{ij} \in \mathcal{E}$, the MPNNs operations for each layer $l = 0, \ldots, L-1$ can be summarised as:

$$\textbf{message+aggregate:} \quad \boldsymbol{m}_i^{l+1} = \bigoplus_{j \in \mathcal{N}(i)} M_l(\boldsymbol{h}_j^l, a_{ij})$$

$$\textbf{update:} \quad \boldsymbol{h}_i^{l+1} = U_l(\boldsymbol{h}_i^l, \boldsymbol{m}_i^{l+1}) \tag{1}$$

where $\boldsymbol{h}_i^l$ is the feature vector for node $i$ at layer $l$, $a_{ij}$ are the edge attributes, and $\bigoplus_{j \in \mathcal{N}(i)}$ is a permutation invariant pooling operation over the node $i$ neighbors $\mathcal{N}(i)$. Finally, $M_l, U_l$ are learnable message and update functions usually approximated with small neural networks.

## 2.2. Machine Learning Interatomic Potentials

Density Functional Theory (DFT) (Hohenberg and Kohn, 1964; Kohn and Sham, 1965) is a widely used quantum mechanical method for calculating a range of properties for various atomic systems. Under the Born-Oppenheimer approximation (Born and Heisenberg, 1985), the potential energy surface (PES) $E$ of an atomic system depends only on positions $r$ and atomic numbers $Z$, with forces given by $F = -\nabla_r E(r, Z)$. However, DFT calculations are computationally expensive due to their cubic scaling with respect to the number of electrons; therefore, Machine Learning Interatomic Potentials (MLIPs) based on (equivariant) MPNNs (Batatia et al., 2022; Fu et al., 2025; Neumann et al., 2024; Rhodes et al., 2025; Schütt et al., 2017) have emerged as efficient surrogates showing remarkable success, particularly due to linear scaling with system size and ability to generalise across chemical domains (Wood et al., 2025). Despite these advances, MLIPs often fail to generalise to out-of-distribution data, such as out-of-equilibrium structures or rare chemical compositions, highlighting the need for principled UQ.

## 2.3. Uncertainty Quantification in Machine Learning Interatomic Potentials

Uncertainty quantification in MLIPs plays a central role in ensuring model reliability (Thaler et al., 2023; Tan et al., 2023; Chmiela et al., 2019; Kreiman and Krishnapriyan, 2025), as well as in guiding efficient exploration of chemical space through active learning (Kulichenko et al., 2024; Jinnouchi et al., 2020; Zaverkin et al., 2024). Among existing approaches, Deep Ensembles MLIPs (Tan et al., 2023; van der Oord et al., 2023; Kulichenko et al., 2023) remain one of the most effective UQ methods in terms of both predictive accuracy and uncertainty estimation. However, Deep Ensembles are computationally expensive, with training and inference costs scaling linearly with ensemble size (one model needs to be trained/queried for each ensemble member). Moreover, they are known to exhibit calibration issues (Pernot, 2022), often requiring post-hoc calibration techniques (Hu et al., 2022). Finally, for large-scale foundation models, constructing and maintaining multiple ensemble members becomes typically infeasible due to substantial computational and storage demands, leaving only a single pretrained model available for downstream tasks. For these reasons, we present a Bayesian approach that is scalable, requiring only a single trained model, and empirically exhibits good calibration, offering a practical and efficient solution for UQ in MLIPs.

## 3. Methods

BLIP is a practical and easy-to-use approach for training or fine-tuning any MLIP using variational Bayesian infer-

ence. It treats the weights of the message and update functions stochastically by injecting input-dependent Gaussian noise, allowing the model to capture uncertainty that varies with the atomic input structure. In the next section, we describe how BLIP can be seamlessly integrated into standard MLIP architectures, trained across diverse tasks, and scaled to support large models.

### 3.1. Building Bayesian Interatomic Potentials

As previously mentioned, MLIPs are typically implemented using MPNNs, where both the message and update functions are parameterised by separate neural networks. Following the Bayesian paradigm, we model the message and update functions $M_l$ and $U_l$ at each layer $l$ as Bayesian neural networks. Specifically, these functions are parameterized by random weights $\boldsymbol{\omega}_l^M$ and $\boldsymbol{\omega}_l^U$, drawn independently from prior distributions $p(\boldsymbol{\omega}_l^M, \boldsymbol{\omega}_l^U) = p(\boldsymbol{\omega}_l^M)p(\boldsymbol{\omega}_l^U)$. This formulation enables the message passing process to incorporate uncertainty in the parameters of both the message and update functions, allowing uncertainty to propagate across layers.

Assuming a complete factorisation across layers, the generative model for the hidden state $\boldsymbol{h}_i^{l+1}$ at layer $l+1$ is given by:

$$p(\boldsymbol{h}_i^{l+1}, \boldsymbol{h}_i^l, \boldsymbol{\omega}_l^M, \boldsymbol{\omega}_l^U) = p(\boldsymbol{h}_i^{l+1} \mid \boldsymbol{h}_i^l, \boldsymbol{\omega}_l^M, \boldsymbol{\omega}_l^U) \\ \times p(\boldsymbol{\omega}_l^M)\, p(\boldsymbol{\omega}_l^U). \tag{2}$$

where the conditional distribution is the pushforward induced by the standard message passing procedure:

$$\boldsymbol{m}_i^{l+1} = \bigoplus_{j \in \mathcal{N}(i)} M_l(\boldsymbol{h}_j^l, a_{ij}; \boldsymbol{\omega}_l^M) \\ \boldsymbol{h}_i^{l+1} = U_l(\boldsymbol{h}_i^l, \boldsymbol{m}_i^{l+1}; \boldsymbol{\omega}_l^U) \tag{3}$$

which is a deterministic delta distribution:

$$p(\boldsymbol{h}_i^{l+1} \mid \boldsymbol{h}_i^l, \boldsymbol{\omega}_l^M, \boldsymbol{\omega}_l^U) = \delta(\boldsymbol{h}_i^{l+1} - g(\boldsymbol{h}_i^l; \boldsymbol{\omega}_l^M, \boldsymbol{\omega}_l^U)), \tag{4}$$

with $g$ the function obtained with the message and update composition above. In practice, the model works by first sampling random weights for the message and update functions at each layer. Then, using these sampled weights, it performs the usual message-passing updates starting from the initial hidden state. The final output $\mathbf{y}$ comes from the last layer's hidden state $\mathbf{h}^L$, which depends on the initial features, edge attributes, and all sampled weights up to that layer. In other words, $\mathbf{h}^L = f\left(\mathbf{h}^0, a_{ij}, \{\boldsymbol{\omega}_l^M, \boldsymbol{\omega}_l^U\}_{l=0}^{L-1}\right)$.

### 3.2. Variational Adaptive Dropout

**Variational Posterior:** To train the model, we introduce a variational posterior distribution over the weights, conditioned on the input features (initial node representations $\mathbf{h}^0$

and edge attributes $a_{ij}$). This allows the model to capture input-dependent uncertainty more accurately. Formally, we factorise the posterior as:

$$q_{\boldsymbol{\phi}}(\boldsymbol{\omega}_l^M, \boldsymbol{\omega}_l^U \mid \mathbf{h}^0, a_{ij}) = q_{\boldsymbol{\phi}}(\boldsymbol{\omega}_l^M \mid \mathbf{h}^0, a_{ij})\, q_{\boldsymbol{\phi}}(\boldsymbol{\omega}_l^U \mid \mathbf{h}^0), \tag{5}$$

where $\boldsymbol{\phi}$ denotes the parameters of the approximate distribution. Following Coscia et al. (2025), we employ an efficient parameterisation based on Variational Dropout (Kingma et al., 2015), which enables scalable inference in large MPNN architectures thanks to the local-reparametrization trick (Kingma et al., 2015) (see Algorithm 1). Specifically, in each message-passing layer $l$, the message and update functions can be parametrised by a multi-layer perceptron (MLP), which may consist of multiple successive linear layers. For example, the message function at layer $l$ could be represented as an MLP with $S$ linear layers. We model the weights of each such internal linear layer $s = 1, \ldots, S$ using a Gaussian distribution[1]:

$$\begin{aligned} q_{\boldsymbol{\phi}}(\boldsymbol{\omega}_{ls}^M \mid \mathbf{h}^0, a_{ij}) &= \mathcal{N}\left(\boldsymbol{\theta}_{ls}^M, \alpha_l(\boldsymbol{\theta}_{ls}^M)^2\right), \\ q_{\boldsymbol{\phi}}(\boldsymbol{\omega}_{ls}^U \mid \mathbf{h}^0) &= \mathcal{N}\left(\boldsymbol{\theta}_{ls}^U, \beta_l(\boldsymbol{\theta}_{ls}^U)^2\right), \end{aligned} \tag{6}$$

where $\boldsymbol{\theta}_{ls}^M$ and $\boldsymbol{\theta}_{ls}^U$ are the mean weights of the message and update networks, respectively, and the squaring is applied element-wise. The *variational adaptive dropout* coefficients $\alpha_l$ and $\beta_l$ are input-dependent:

$$\alpha_l = \alpha_l(\mathbf{h}^0, a_{ij}) \in \mathbb{R}^+, \quad \beta_l = \beta_l(\mathbf{h}^0) \in \mathbb{R}^+, \tag{7}$$

and are computed per edge and node, respectively by first obtaining a number $r \in [0, 1]$ and then applying the transformation $\frac{r}{1-r}$. This can be interpreted as injecting Gaussian noise into the weights, scaled by the corresponding variational dropout coefficients:

$$\begin{aligned} \boldsymbol{\omega}_{ls}^M &= \boldsymbol{\theta}_{ls}^M + \boldsymbol{\epsilon}_{ls}^M \odot |\boldsymbol{\theta}_{ls}^M|, \quad \boldsymbol{\epsilon}_{ls}^M \sim \mathcal{N}\left(0, \alpha_l(\mathbf{h}^0, a_{ij})\right), \\ \boldsymbol{\omega}_{ls}^U &= \boldsymbol{\theta}_{ls}^U + \boldsymbol{\epsilon}_{ls}^U \odot |\boldsymbol{\theta}_{ls}^U|, \quad \boldsymbol{\epsilon}_{ls}^U \sim \mathcal{N}\left(0, \beta_l(\mathbf{h}^0)\right), \end{aligned} \tag{8}$$

where $\odot$ denotes element-wise multiplication. Due to the Gaussian assumption, the local reparameterization trick (Kingma et al., 2015) can be employed, allowing us to sample activations directly instead of weights. This drastically reduces the number of random samples per iteration, as we only need to sample once for every neuron activation rather than once for every weight, leading to both lower variance in the gradient estimates and improved computational efficiency (see Appendix D).

**Encoding Variational Adaptive Dropout Coefficients** The adaptive variance terms $\alpha_l$ and $\beta_l$ are generated by a

[1]We assume the same number of internal layers for both message and update networks for notational simplicity, though the approach trivially generalises. We also factorise across this dimension.

dedicated inference network $E_{\boldsymbol{\psi}}$, with two separate output heads. This inference network is shared across all layers $l$, meaning that the coefficients are computed once—based solely on the initial node features $\mathbf{h}^0$ and edge attributes $a_{ij}$, and then broadcast to each message-passing layer. As a result, the full set of trainable variational parameters $\boldsymbol{\phi}$ consists of: (i) the unperturbed weights of the message-passing network, $\boldsymbol{\Theta} = \{\boldsymbol{\theta}_l^M, \boldsymbol{\theta}_l^U\}_{l=0}^{L-1}$; and (ii) the inference network parameters $\boldsymbol{\psi}$, which govern the input-dependent noise injected into each layer.

**Evidence Lower Bound Optimisation** We optimise the variational parameters $\boldsymbol{\phi}$ by maximising the Evidence Lower Bound (ELBO), which trades off between data fit (the expected log-likelihood term) and model complexity (KL divergence between the posterior and the prior). The ELBO is given by:

$$\begin{aligned} \mathcal{L}(\boldsymbol{\phi}) = &\, \mathbb{E}_{q_{\boldsymbol{\phi}}}\left[\log p(\mathbf{y} \mid \mathbf{h}^0, a_{ij}, \{\boldsymbol{\omega}_l^M, \boldsymbol{\omega}_l^U\}_{l=0}^{L-1})\right] \\ &- \lambda \sum_{l=0}^{L-1} \Big( D_{\mathrm{KL}}\big[q_{\boldsymbol{\phi}}(\boldsymbol{\omega}_l^M \mid \mathbf{h}^0, a_{ij}) \,\|\, p(\boldsymbol{\omega}_l^M)\big] \\ &\qquad\qquad + D_{\mathrm{KL}}\big[q_{\boldsymbol{\phi}}(\boldsymbol{\omega}_l^U \mid \mathbf{h}^0) \,\|\, p(\boldsymbol{\omega}_l^U)\big]\Big). \end{aligned} \tag{9}$$

where the expectation is taken over all stochastic weights $\{\boldsymbol{\omega}_l^M, \boldsymbol{\omega}_l^U\}_{l=0}^{L-1}$, and $\lambda$ is the KL weighting, as normally done in variational inference optimisation (Higgins et al., 2017). In practice, we approximate the expectation using a single Monte Carlo sample from the variational posterior at each training iteration.

We adopt a zero-mean Gaussian prior with fixed layerwise variance, chosen to match the variance induced by dropout with probability $p \in [0, 1]$:

$$\begin{aligned} p(\boldsymbol{\omega}_{ls}^M) &= \mathcal{N}\left(\mathbf{0}, \frac{p}{1-p}(\boldsymbol{\theta}_{ls}^M)^2\right), \\ p(\boldsymbol{\omega}_{ls}^U) &= \mathcal{N}\left(\mathbf{0}, \frac{p}{1-p}(\boldsymbol{\theta}_{ls}^U)^2\right), \end{aligned} \tag{10}$$

for each message-passing layer $l$ and internal linear layer $s$. This choice yields closed-form KL divergences for both terms in equation 9, as reported in Appendix A.1. Algorithm 2 shows how to train MLIPs using BLIP.

### 3.3. Equivariance and Invariance Analysis

Many MLIP architectures are explicitly designed to be either *equivariant* or *invariant* to certain input transformations (Cohen and Welling, 2016; Weiler et al., 2023; Bronstein et al., 2017; Cohen et al., 2019). An *equivariant* network ensures that when the input is transformed (e.g., via translation or rotation), the output transforms accordingly. An *invariant* network is a special case of an equivariant one, where it produces the same output regardless of such

*Table 1.* Results for the N-body system. Averages are computed over 4 random initialisation seeds.

| Equivariant | Model | MSE $\times 10^{-1}$ ($\downarrow$) | NLL ($\downarrow$) | CRPS ($\downarrow$) |
|---|---|---|---|---|
| ✗ | GNN (base model) | $0.121^{\pm 0.004}$ | - | - |
| | MC Dropout ($p = 0.2$) | $0.117^{\pm 0.001}$ | $-7.83^{\pm 0.31}$ | $0.043^{\pm 0.000}$ |
| | MC Dropout ($p = 0.5$) | $0.138^{\pm 0.049}$ | $-8.03^{\pm 2.13}$ | $0.046^{\pm 0.011}$ |
| | Deep Ensemble | $0.106^{\pm \text{NA}}$ | $-3.89^{\pm \text{NA}}$ | $0.034^{\pm \text{NA}}$ |
| | BLIP (ours) | $\mathbf{0.092}^{\pm 0.004}$ | $\mathbf{-9.03}^{\pm 0.50}$ | $\mathbf{0.032}^{\pm 0.001}$ |
| ✓ | EGNN (base model) | $0.069^{\pm 0.002}$ | - | - |
| | MC Dropout ($p = 0.2$) | $0.099^{\pm 0.006}$ | $-6.19^{\pm 0.42}$ | $0.044^{\pm 0.006}$ |
| | MC Dropout ($p = 0.5$) | $0.299^{\pm 0.011}$ | $0.91^{\pm 0.26}$ | $0.091^{\pm 0.001}$ |
| | Deep Ensemble | $\mathbf{0.057}^{\pm \text{NA}}$ | $\mathbf{-16.22}^{\pm \text{NA}}$ | $\mathbf{0.020}^{\pm \text{NA}}$ |
| | BLIP (ours) | $0.060^{\pm 0.003}$ | $-12.52^{\pm 0.72}$ | $0.022^{\pm 0.001}$ |

transformations. These properties are crucial when modelling atomic systems, where geometric symmetries should not alter the model's predictions. BLIP can also be used in conjunction with *equivariant* or *invariant* MLIPs without violating their symmetry properties[2]. To ensure this, the variational dropout coefficients $\alpha$ and $\beta$ must themselves be invariant, which can be achieved by implementing the inference network $E_\psi$ as an invariant neural network. When $\alpha$ and $\beta$ are invariant to input transformations, the local reparameterization trick can be safely applied to any equivariant networks whose learnable layers can be expressed as unconstrained linear transformations without an irreducible representation (irrep) dimension. This includes models such as those in (Satorras et al., 2021; Schütt et al., 2021; Köhler et al., 2019; Schütt et al., 2017). However, in its current form, it excludes other important classes of equivariant MLIPs, such as MACE (Batatia et al., 2022) and UMA (Wood et al., 2025), thereby motivating future work to bridge this gap. A more in-depth discussion is given in Appendix A.3.

## 4. Experiments

We demonstrate the effectiveness of BLIP across a range of simulation-based chemistry tasks. As a first illustrative example, we use BLIP to model a system of charged particles interacting via Coulomb forces. We then apply BLIP to three challenging MLIP tasks: (i) training from scratch using the equivariant PaiNN architecture (Schütt et al., 2021) on a data-scarce, out-of-distribution ammonia ($NH_3$) system, and fine-tuning the pretrained ORB v3 model (Rhodes et al., 2025) on (ii) the silica glass ($SiO_2$) with 699 atoms,

---

[2]Although the model is equivariant in distribution, individual forward passes may break exact equivariance or invariance due to the injected noise. For a given input, different noise samples (which themselves do not need to be equivariant) can yield different outputs. However, when averaged across samples, the statistical moments remain equivariant.

and (iii) the MATPES (Kaplan et al., 2025) PBE dataset using active learning.

### 4.1. Charged Particle System

We use the Coloumbs' N-body particle benchmark (Fuchs et al., 2020) as an initial example to test BLIP's uncertainty quantification abilities. The system consists of 5 charged (positive or negative) particles, each equipped with a position and velocity in 3D space. The task is to predict the particle positions after 1000 timesteps, given their initial positions and velocities. We use the same split for training and testing as Satorras et al. (2021), and trained the models using the standard mean squared error loss, augmented with a KL divergence term for BLIP. Further dataset and metrics details are provided in Section B.

We benchmark two architectures for this task: a standard Graph Neural Network (GNN) (Gilmer et al., 2017) and the Equivariant Graph Neural Network (EGNN) proposed by (Satorras et al., 2021) using the same network hyperparameters and training protocol as in (Satorras et al., 2021). We then evaluate two Bayesian versions of the same architectures for UQ: Monte Carlo (MC) Dropout (Gal and Ghahramani, 2016), using different dropout probabilities $p$; and Deep Ensembles (Lakshminarayanan et al., 2017) with four ensemble members. For BLIP, a compact invariant posterior network for both the message and update function coefficients is used. Additional architecture and training details are provided in Section C.

**Results:** Table 1 reports the predictive performance and uncertainty quantification metrics on the particle system, averaged over four random seeds. Our method, BLIP, consistently outperforms MC Dropout at different dropout probabilities, and matches or exceeds Deep Ensembles in terms of both accuracy and uncertainty quality. In the non-equivariant setting (GNN), BLIP achieves the lowest MSE and best-calibrated uncertainty estimates, as indi-

cated by the lowest NLL and CRPS. In the equivariant setting (EGNN), Deep Ensembles obtain the best overall performance, but only marginally surpass BLIP. Importantly, BLIP remains highly competitive while being significantly more efficient, requiring only a single model for both training and inference against four ensemble members for Deep Ensemble ($\times 4$ main model parameter size). Overall, these results demonstrate that BLIP offers high-quality uncertainty estimates without compromising predictive accuracy, and does so with lower computational overhead than common ensemble-based methods.

## 4.2. Learning Interatomic Potentials in Data-Scarce and Out-Of-Distribution Regimes

This experiment focuses on the study of the Ammonia ($NH_3$) molecule for assessing the model's robustness in a realistic low-data, high out-of-distribution (OOD) regime. This is a common challenge in materials science, where training data may be biased toward equilibrium or near-equilibrium configurations. We use the Ammonia dataset introduced by (Tan et al., 2023), which consists of a small number of $NH_3$ molecular geometries sampled across a range of energy configurations. The training set includes 78 geometries corresponding to near-equilibrium configurations with energies in the range of 0 to 5 kcal/mol. The test set, in contrast, contains 129 OOD geometries with energies spanning up to 70 kcal/mol, thereby evaluating model generalisation far from the training distribution. We evaluate performance using three metrics: mean absolute error (MAE), expected calibration error (ECE), and the Spearman correlation. Further dataset and metrics details are provided in Section B.

We adopt the (equivariant) PaiNN architecture (Schütt et al., 2021) as our machine learning interatomic potential. To ensure a fair comparison, we use the same hyperparameter settings, training protocol, and evaluation procedure as described in (Tan et al., 2023). We train the model by explicitly enforcing energy conservation, computing atomic forces as the negative gradient of the predicted potential energy with respect to atomic positions, and weighting the energy/forces terms accordingly to Tan et al. (2023). For BLIP, a compact invariant posterior network for both the message and update function coefficients is used. Additional architecture and training details are provided in Section C.

**Results:** As shown in Table 2, BLIP achieves the best MAE performance across both energy and forces targets, outperforming all baseline models from (Tan et al., 2023). For energy prediction, BLIP achieves an MAE of $11.51 \pm 0.75$, slightly better than the Deep Ensemble (11.71) and substantially better than other probabilistic methods. On force prediction, which is a more challenging task, BLIP

*Table 2.* Energy and Forces mean absolute error in the Ammonia system for OOD data. Averages are computed over 4 random initialisation seeds. Baseline models' results are from (Tan et al., 2023).

| Model | MAE ($\downarrow$) | |
|---|---|---|
| | Energy [kcal/mol] | Forces [kcal/mol/Å] |
| Deep Ensemble | 11.71 | 48.03 |
| Evidential Regression | 13.58 | 53.29 |
| Mean Variance Estimate | 13.47 | 53.56 |
| Gaussian Mixture Model | 12.51 | 50.17 |
| BLIP (ours) | $\mathbf{11.51}^{\pm 0.75}$ | $\mathbf{41.93}^{\pm 3.38}$ |

shows a notable improvement with an MAE of $41.93 \pm 3.38$, significantly lower than both the Deep Ensemble and other baselines. These results demonstrate that training an MLIP with BLIP yields superior predictive accuracy on both energy and force targets, without modifying the base architecture or requiring multiple models, as is the case with Deep Ensembles. Regarding uncertainty estimation, Table 3 reports the expected calibration error for the different models. ECE measures how well the estimated probabilities match the observed probabilities, which intuitively means that if a model predicts to be confident $n\%$ of the time, it should be correct approximately $n\%$ of the time. From the table, it is clear that BLIP is the most calibrated model, outperforming even the Deep Ensemble method. Additionally, we analyse the linear relationship between predicted uncertainty (variance) and squared error (Figure 2), which is particularly relevant for active learning. Our model demonstrates a strong linear correlation (Spearman correlation of $0.863 \pm 0.024$), indicating its reliability in identifying structures where the MLIP predictions are likely inaccurate.

## 4.3. Finetuning Pretrained Interatomic Potentials on $SiO_2$

This experiment focuses on fine-tuning a pretrained MLIP on $SiO_2$. In the current era of increasing data and compute scale, training universal models from scratch is often impractical without large computational resources. Instead, it is common to fine-tune existing pretrained models. Among recent advancements, we select the ORB v3 Direct 20OMat model (Rhodes et al., 2025), a state-of-the-art direct force field that is not rotationally equivariant. We fine-tune the silica glass ($SiO_2$) dataset introduced by (Tan et al., 2023), which consists of system snapshots extracted from diverse, long molecular dynamics trajectories. Each configuration contains 699 atoms in total, including 233 silicon and 466 oxygen atoms. This dataset presents a realistic and challenging benchmark, as the formation of large bulk struc-

*Table 3.* Expected calibration error for Forces in the Ammonia system for OOD data. Averages are computed over 4 random initialisation seeds. Baseline models' results are from (Tan et al., 2023).

| Model | ECE ($\downarrow$) | $\rho$ ($\uparrow$) |
|---|---|---|
| Deep Ensemble | 0.108 | 0.825 |
| Evidential Regression | 0.120 | 0.779 |
| Mean Variance Estimate | 0.135 | 0.693 |
| Gaussian Mixture Model | 0.112 | **0.904** |
| BLIP (ours) | **0.091**$^{\pm 0.006}$ | 0.863$^{\pm 0.024}$ |

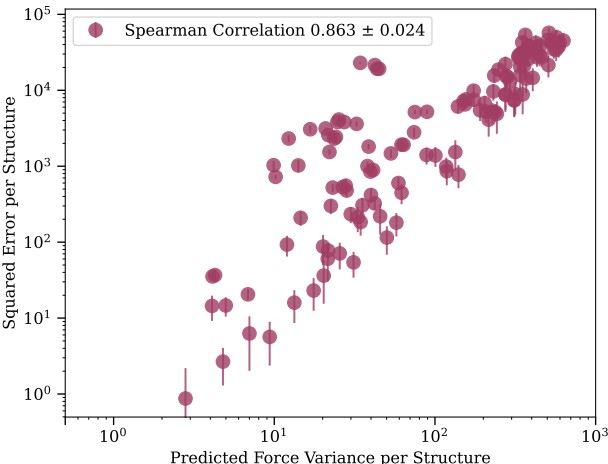

*Figure 2.* Spearman correlation between predicted uncertainty and squared mean error for Forces [kcal/mol/Å] in the Ammonia system for OOD data.

tures makes DFT calculations computationally expensive. We fine-tune models using varying numbers of training samples (32, 128, and 1024) and evaluate performance on a held-out test set of 373 samples. We report performance using three metrics: mean absolute error, expected calibration error, and Spearman correlation. Further details on the dataset and evaluation metrics can be found in Section B.

We fine-tune the non-conservative ORB v3 model, originally trained on a subset of ab-initio molecular dynamics data from OMAT 24 (Barroso-Luque et al., 2024). We select ORB v3 due to its strong performance among state-of-the-art MLIPs and its low inference latency, benefiting from its non-conservative nature, which eliminates the need for automatic differentiation during force computation. We fine-tuned ORB v3 both with (**w/**) and without (**w/o**) BLIP. We also fine-tuned a Deep Ensemble of 4 members for direct UQ comparison. Additional architectural and training details are provided in Section C.

**Results:** Figure 3 shows the results of fine-tuning the ORB v3 model on the Silica dataset. For comparison, we include in the caption the best-performing model reported in the literature on this dataset, a Deep Ensemble Equivari-

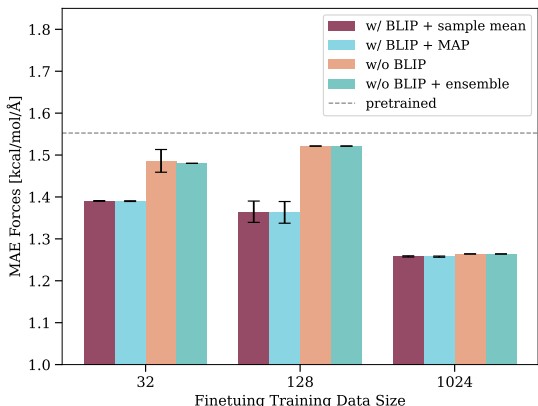

*Figure 3.* Forces mean absolute error in the Silica system across different training sizes for finetuned ORB v3 direct 2OOMAT. Averages are computed over 4 random initialisation seeds, and the standard error of the mean is reported as an error bar. The best baseline model is a Deep Ensemble Equivariant PaiNN from (Tan et al., 2023) trained on the 1024 configuration (same split), achieving MAE 6.58 kcal/mol/Å.

ant PaiNN (4 members) from (Tan et al., 2023). Notably, this model yields a substantially higher error than even the pretrained ORB v3 model (**pretrained**), highlighting the potential advantages of universal MLIPs. For inference with the BLIP fine-tuned models, we evaluated predicted forces using two approaches: (i) the sample mean computed from 100 stochastic forward passes (**sample mean**, Algorithm 3), and (ii) a single forward pass using the unperturbed weights, i.e., the maximum a posteriori estimate (**MAP estimate**, Algorithm 4).

The results demonstrate that BLIP is both accurate and efficient. Using the sample mean or the more efficient MAP estimate, BLIP outperforms all ORB v3 baseline models and the pretrained PaiNN ensemble. Remarkably, BLIP with MAP inference achieves better accuracy than deepensembling the fine-tuned models, at a fraction of the computational cost, since it requires storing and fine-tuning only a single model. For uncertainty quantification, we evaluate ECE and Spearman correlation on the 1024-sample setup. BLIP achieves an ECE of $0.032 \pm 0.002$, while Deep Ensemble performs significantly worse with $0.247$. When examining the correlation between predicted variance and squared error, crucial for active learning, BLIP slightly underperform, with a Spearman correlation of $0.602 \pm 0.002$ versus $0.676$ for Deep Ensemble. Overall, the results highlight that BLIP provides accurate calibration and reliable uncertainty estimates for guiding model refinement. More importantly, fine-tuning with BLIP achieves higher predictive accuracy than both deterministic and ensemble baselines, while maintaining the same inference cost as a standard deterministic model.

*Table 4.* Energy and Forces mean absolute error in the MATPES PBE dataset. Averages and standard deviations are computed over 3 random initialization seeds.

| Model | MAE ($\downarrow$) | |
|---|---|---|
| | Energy [meV/atom] | Forces [meV/Å] |
| w/o BLIP | $111.83^{\pm 1.24}$ | $110.84^{\pm 0.57}$ |
| w/ BLIP + random | $96.65^{\pm 0.09}$ | $105.12^{\pm 0.06}$ |
| w/ BLIP + uncertainties | $\mathbf{91.13}^{\pm 1.14}$ | $\mathbf{103.85}^{\pm 0.14}$ |

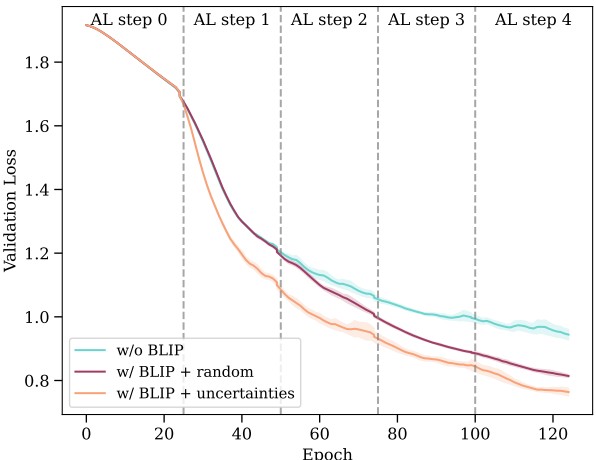

*Figure 4.* Validation loss for different models during each active learning step, calculated as the sum of mean absolute errors for energy and forces. Results are averaged over 3 independent training runs. Energy is reported in eV and forces in eV/Å.

### 4.4. Active Learning Fine-Tuning of Pretrained Interatomic Potentials

In this experiment, we fine-tune the ORB v3 Direct 20OMat model on the MATPES PBE dataset (Kaplan et al., 2025), which comprises roughly 400,000 carefully sampled structures from 281 million molecular dynamics snapshots, covering 16 billion atomic environments. To efficiently improve model accuracy, we employ an Active Learning (AL) strategy. In this framework, BLIP uncertainties are used to identify structures where the pretrained model is most uncertain. These high-uncertainty structures are then computed at the reference DFT level and used to iteratively fine-tune the model. By repeating this cycle, the model progressively learns from the most informative data points, maximizing performance gains while minimizing computational cost. We assess model performance using the mean absolute error for both forces and energies, and we track the evolution of the validation loss throughout the AL iterations. Detailed descriptions of the dataset and evaluation metrics are provided in Section B, while implementation details are provided in Section C.

**Results:** The validation loss in Figure 4 exhibits a clear and consistent order of convergence across models. Fine-tuning with BLIP using uncertainty-guided selection of structures (w/ BLIP + uncertainties), where structures are chosen based on the variance in predicted energies from a pool, achieves the fastest and most stable convergence to lower error. Fine-tuning with BLIP on randomly selected structures (w/ BLIP + random) converges to a worse optimum, but still outperforms the model without BLIP (w/o BLIP), which shows the poorest performance. Importantly, additional active learning steps further amplify the performance gap between uncertainty-guided and random selection strategies. This trend is also reflected in the test errors for both forces and energies, which follow the same ordering (Table 4). These results demonstrate that fine-tuning with BLIP consistently improves model quality and that leveraging BLIP uncertainties for active learning not only accelerates convergence but also systematically enhances predictive accuracy.

## 5. Conclusions

We introduced BLIP, a scalable and architecture-agnostic variational Bayesian framework for principled uncertainty quantification in (equivariant) message passing neural networks. BLIP can be used for both training from scratch and fine-tuning pretrained MPNNs. Unlike deep ensembles, it maintains the same inference-time efficiency as standard deterministic models for predictive tasks, while enabling well-calibrated uncertainty estimates through stochastic sampling. We demonstrated the effectiveness of BLIP across a range of scientific tasks, including particle prediction and machine-learned interatomic potentials. BLIP achieves remarkable performance in prediction and uncertainty quantification for MLIPs when trained in data-scarce and OOD regimes. Notably, we showed that fine-tuning pretrained MLIPs with BLIP consistently enhances model performance, outperforming both deterministic and ensemble-based baselines. In addition, the uncertainty estimates produced by BLIP are highly calibrated and strongly correlated with model error, making them particularly useful for downstream applications such as active learning and error detection. These results highlight the potential of BLIP to serve as a drop-in tool for improving the performance of MLIPs, more generally MPNNs, and provide them with useful uncertainty estimates.

## Impact Statement

This paper presents work aimed at advancing the field of machine learning for atomistic modeling, with the potential to accelerate scientific discovery in chemistry and the biological sciences. As machine learning–based atomistic modeling continues to evolve, we recognize that there are

numerous potential societal consequences of this work. While we do not feel that any specific issues need to be highlighted, we emphasize the importance of ongoing discussions on ethical considerations and best practices to ensure these technologies are developed and applied in ways that benefit society as a whole.

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

# APPENDIX

## TABLE OF CONTENTS

# A. Proofs and Derivations

This Appendix Section is devoted to the formal proofs introduced in the main text.

## A.1. KL-divergence proof

We want to compute $D_{\mathrm{KL}}^{\mathrm{tot}} = D_{\mathrm{KL}}\big[q_\phi(\boldsymbol{\omega}^M \mid \mathbf{h}^0, a_{ij}) \,\|\, p(\boldsymbol{\omega}^M)\big] + D_{\mathrm{KL}}\big[q_\phi(\boldsymbol{\omega}^U \mid \mathbf{h}^0) \,\|\, p(\boldsymbol{\omega}^U)\big]$. Due to factorisation on the layers, the KL divergence reads:

$$D_{\mathrm{KL}}^{\mathrm{tot}} = \sum_{l=1}^{L} \sum_{s=1}^{S} \big[ D_{\mathrm{KL}}\big[q_\phi(\boldsymbol{\omega}_{ls}^M \mid \mathbf{h}^0, a_{ij}) \,\|\, p(\boldsymbol{\omega}_l^M)\big] + D_{\mathrm{KL}}\big[q_\phi(\boldsymbol{\omega}_{ls}^U \mid \mathbf{h}^0) \,\|\, p(\boldsymbol{\omega}_{ls}^U)\big]\big] \tag{11}$$

For the variational posterior and prior defined as in the main text:

$$\begin{aligned}
q_\phi(\boldsymbol{\omega}_{ls}^M \mid \mathbf{h}^0, a_{ij}) &= \mathcal{N}\left(\boldsymbol{\theta}_{ls}^M, \, \alpha_l(\boldsymbol{\theta}_{ls}^M)^2\right), \quad p(\boldsymbol{\omega}_{ls}^M) = \mathcal{N}\left(0, \, \frac{p}{1-p}(\boldsymbol{\theta}_{ls}^M)^2\right), \\
q_\phi(\boldsymbol{\omega}_{ls}^U \mid \mathbf{h}^0) &= \mathcal{N}\left(\boldsymbol{\theta}_{ls}^U, \, \beta_l(\boldsymbol{\theta}_{ls}^U)^2\right), \quad p(\boldsymbol{\omega}_{ls}^U) = \mathcal{N}\left(0, \, \frac{p}{1-p}(\boldsymbol{\theta}_{ls}^U)^2\right),
\end{aligned} \tag{12}$$

the marginal divergences are analytical:

$$\begin{aligned}
D_{\mathrm{KL}}\big[q_\phi(\boldsymbol{\omega}_{ls}^M \mid \mathbf{h}^0, a_{ij}) \,\|\, p(\boldsymbol{\omega}_l^M)\big] &= \frac{1}{2}\left(\frac{(\alpha_l + 1)(1-p)}{p} + \log\left(\frac{p}{1-p}\right) - \log(\alpha_l) - 1\right), \\
D_{\mathrm{KL}}\big[q_\phi(\boldsymbol{\omega}_{ls}^U \mid \mathbf{h}^0, a_{ij}) \,\|\, p(\boldsymbol{\omega}_l^U)\big] &= \frac{1}{2}\left(\frac{(\beta_l + 1)(1-p)}{p} + \log\left(\frac{p}{1-p}\right) - \log(\beta_l) - 1\right).
\end{aligned} \tag{13}$$

In practice, we weight the KL term by a factor $\lambda$ as normally done for ELBO optimisation (Alemi et al., 2018; Higgins et al., 2017).

## A.2. Predictive Distribution and Uncertainty Estimates

This subsection shows how to use BLIP for inference and uncertainty quantification. In the inference case, we are interested in regressing the output from the input by marginalising over the weights, while in UQ we aim to model epistemic (model weight uncertainty) and/or aleatory uncertainty (data's inherent randomness).

**Predictive Distribution**    The model *predictive distribution* $p\left(\boldsymbol{y} \mid \boldsymbol{h}^L = f\left(\mathbf{h}^0, a_{ij}, \{\boldsymbol{\omega}_l^M, \boldsymbol{\omega}_l^U\}\right)\right)$ is obtained by marginalizing over the weights $\boldsymbol{\omega}$ sampled from the posterior distribution:

$$p(\boldsymbol{y} \mid \boldsymbol{h}^L) = \mathbb{E}_{q_\phi}\left[p\left(\boldsymbol{y} \mid \boldsymbol{h}^L = f\left(\mathbf{h}^0, a_{ij}, \{\boldsymbol{\omega}_l^M, \boldsymbol{\omega}_l^U\}\right)\right)\right]. \tag{14}$$

In practice, this corresponds to performing multiple stochastic forward passes through the network and averaging the outputs, following the standard Monte Carlo approximation of predictive distributions (Gal and Ghahramani, 2016). Alternatively, a deterministic approximation can be used by evaluating the network at the posterior mean weights when uncertainty is not needed.

**Uncertainty Estimates**    The predictive distribution allows for a clear variance decomposition using the law of total variance:

$$\boldsymbol{\sigma}^2 = \underbrace{\boldsymbol{\sigma}_{\boldsymbol{\omega}}^2(\mathbb{E}[\boldsymbol{y} \mid \boldsymbol{h}^L])}_{\text{epistemic uncertainty}} + \underbrace{\mathbb{E}_{\boldsymbol{\omega}}[\boldsymbol{\sigma}^2(\boldsymbol{y} \mid \boldsymbol{h}^L)]}_{\text{aleatoric uncertainty}}, \tag{15}$$

with $\mathbb{E}[\boldsymbol{y} \mid \boldsymbol{h}^L], \boldsymbol{\sigma}^2(\boldsymbol{y} \mid \boldsymbol{h}^L)$ the mean and variance of the distribution $p(\boldsymbol{y} \mid \boldsymbol{h}^L)$ respectively.

## A.3. BLIP Stochastic Equivariance and Local Reparametrization

This section is dedicated to the formal proofs of the equivariance in the BLIP model. We first provide a general proposition for Stochastic Equivariant Maps, and we later connect this to the BLIP update rule.

**Stochastic Equivariant Maps:** Let $G$ be a group acting measurably on spaces $X$ and $Y$. Let $g \cdot x \in X$, and $g \cdot y \in Y$ denote the respective actions. Assume:

1. An invariant map $\mathrm{inv} : X \to \mathrm{Inv}(X)$ such that

$$\mathrm{inv}(g \cdot x) = \mathrm{inv}(x) \quad \forall g \in G, \, x \in X.$$

2. A class $\mathcal{F}$ of deterministic equivariant maps $f : X \to Y$, i.e.

$$f(g \cdot x) = g \cdot f(x) \quad \forall f \in \mathcal{F}, \, g \in G, \, x \in X.$$

3. A measurable map

$$E : \mathrm{Inv}(X) \to \mathcal{P}(\mathcal{F})$$

assigning to each invariant value a probability distribution over equivariant maps.

Define a stochastic map:

$$P[y \mid x] = \int_{\mathcal{F}} \delta(y - f(x)) dE(\mathrm{inv}(x)) \tag{16}$$

i.e. sample $f \sim E(\mathrm{inv}(x))$ and output the deterministic value $f(x)$.

**Proposition 1.** *The stochastic map $P$ is $G$-equivariant:*

$$P[y \mid g \cdot x] = g_* P[y \mid x] \quad \forall g \in G, \, x \in X, \tag{17}$$

*where $g_* : \mathcal{P}(Y) \to \mathcal{P}(Y)$ is the pushforward by the action $y \mapsto g \cdot y$.*

*Proof.* Fix $g \in G$ and $x \in X$. By definition of $P$,

$$P[y \mid g \cdot x] = \int_{\mathcal{F}} \delta\big(y - f(g \cdot x)\big) \, dE(\mathrm{inv}(g \cdot x)). \tag{18}$$

Since $\mathrm{inv}$ is invariant, $\mathrm{inv}(g \cdot x) = \mathrm{inv}(x)$. Thus

$$P[y \mid g \cdot x] = \int_{\mathcal{F}} \delta\big(y - f(g \cdot x)\big) \, dE(\mathrm{inv}(x)). \tag{19}$$

Because each $f \in \mathcal{F}$ is $G$-equivariant, $f(g \cdot x) = g \cdot f(x)$. Substituting,

$$P[y \mid g \cdot x] = \int_{\mathcal{F}} \delta\big(y - g \cdot f(x)\big) \, dE(\mathrm{inv}(x)). \tag{20}$$

Therefore, by the definition of pushforward $g_* \delta(\cdot - f(x)) = \delta(\cdot - g \cdot f(x))$ if follows:

$$P[y \mid g \cdot x] = g_* \left( \int_{\mathcal{F}} \delta\big(y - f(x)\big) \, dE(\mathrm{inv}(x)) \right) = g_* P[y \mid x]. \tag{21}$$

$\square$

**Corollary 1.** *The BLIP update in equation 8 rule applied to an equivariant MLIP yields equivariant stochastic MLIPs.*

To clearly see why Corollary 1 holds, we highlight the following equivalence between the BLIP update and Proposition 1:

- The invariant map $\mathrm{inv}$, is equivalent to the BLIP inference network $E$, mapping the input to the invariant variational dropout coefficients (equation 7).

- The equivariant map $f$ is an equivariant MLIP parametrized by parameters $\theta$.

- The stochastic map $P[y \mid x]$ is the once induced by the BLIP update (equation 8).

It is important to notice that the above corollary applies to any family of equivariant neural networks, including UMA (Wood et al., 2025) and MACE (Batatia et al., 2022). In what follows, we show that the local reparameterization trick (sampling activations) breaks equivariance, whereas perturbing the weights, in principle does not, opening further research to sample activations while remaining equivariant efficiently.

---

**Algorithm 1** *Local reparameterization trick*: Given a minibatch of activations $\boldsymbol{h}$ and probabilistic Gaussian weights $\boldsymbol{\omega} = \boldsymbol{\theta} + \alpha|\boldsymbol{\theta}|\epsilon, \ \epsilon \sim \mathcal{N}(\mathbf{0}, \mathbf{I})$, compute the output of a linear layer using the local reparameterization trick. For the first layer, $\boldsymbol{h}$ corresponds to the state-input.

---

1: $\boldsymbol{m} \leftarrow \texttt{matmul}(\boldsymbol{h}, \boldsymbol{\theta})$
2: $\boldsymbol{std} \leftarrow \alpha \cdot \sqrt{\texttt{matmul}(\boldsymbol{h} \odot \boldsymbol{h}, \boldsymbol{\theta} \odot \boldsymbol{\theta})}$
3: $\boldsymbol{eps} \sim \mathcal{N}(\mathbf{0}, \mathbb{I})$
4: **return** $\boldsymbol{m} + \boldsymbol{std} \odot \boldsymbol{eps}$

---

**Local-reparametrization trick:** The local-reparametrization trick (Kingma et al., 2015) was introduced to sample the predictive distribution of Bayesian linear layers efficiently. For a single linear layer with input $\boldsymbol{h}$ and weight matrix $\boldsymbol{\omega} \sim \mathcal{N}(\boldsymbol{\theta}, \alpha\boldsymbol{\theta}^2)$, the output $\boldsymbol{z}$ also follows a normal distribution:

$$z \sim \mathcal{N}(\boldsymbol{\gamma}, \mathrm{diag}(\boldsymbol{\delta})), \tag{22}$$

where each output element $z_j$ can be written as

$$z_j = \gamma_j + \sqrt{\delta_j}\epsilon_j, \quad \epsilon_j \sim \mathcal{N}(0, 1), \tag{23}$$

with

$$\gamma_j = \sum_i h_i \theta_{ij}, \quad \delta_j = \alpha \sum_i h_i^2 \theta_{ij}^2. \tag{24}$$

This shows that instead of sampling weights $\boldsymbol{\omega}$ directly, one can sample the output $\boldsymbol{z}$ by sampling from a Gaussian with analytically computed mean and variance, which reduces variance and computational cost. In standard deep learning libraries, the above algorithm can be easily implemented by overriding the linear layer module (see Algorithm 1).

**Local-reparametrization trick for equivariant linear maps:** When applying the local reparametrization trick, we push the noise from the weights to the activation, reducing the computational cost. If we apply this to a vector of irreducible dimensions, the trick introduces a different noise scale for each coordinate of the irrep. Since an irrep transforms by mixing its coordinates through a fixed linear representation, all components must be scaled uniformly to preserve equivariance. Local reparam instead applies independent variances to each channel, breaking the required coupling across the irrep. Thus, the resulting stochastic activation is no longer equivariant, despite the underlying weight distribution being equivariant.

## B. Dataset and Metrics

This Appendix Section is devoted to dataset and metrics introduced in the main text.

### B.1. NBody System

**Dataset Generation and Availability:** We utilise the dataset introduced by Fuchs et al. (2020), which consists of simulations of five particles interacting in a three-dimensional box under Coulomb's law. This dataset is publicly available at: https://github.com/FabianFuchsML/se3-transformer-public/tree/master/experiments/nbody/data_generation. Following the setup of Satorras et al. (2021), we generate the dataset using the parameters `--num-train 10000 --seed 43 --sufix small`. We follow the protocol of Satorras et al. (2021) and use 3000 samples each for training, validation, and testing, ensuring that there is no overlap between the splits.

**Metrics:** We adopt three different metrics for prediction and uncertainty estimation. For prediction, we compute the mean squared error (MSE) between the predicted (mean) positions $\hat{\boldsymbol{y}}$ and the true positions $\boldsymbol{y}$:

$$\mathrm{MSE}(\hat{\boldsymbol{y}}, \boldsymbol{y}) = \frac{1}{3N} \sum_{i=1}^{N} \sum_{k=1}^{3} (y_i^k - \hat{y}_i^k)^2, \tag{25}$$

where the index $k$ denotes the spatial dimensions and $i$ indexes the data points.

For uncertainty estimation, we consider the negative log-likelihood (NLL) and the continuous ranked probability score (CRPS). The NLL assumes a Gaussian predictive distribution with mean $\hat{y}$ and variance $\sigma^2$, and is computed as:

$$\text{NLL}(\hat{\boldsymbol{y}}, \boldsymbol{y}, \boldsymbol{\sigma}^2) = \frac{1}{N} \sum_{i=1}^{N} \sum_{k=1}^{3} \left[ \frac{(y_i^k - \hat{y}_i^k)^2}{2\sigma_i^{k\,2}} + \frac{1}{2} \log(2\pi\sigma_i^{k\,2}) \right]. \tag{26}$$

The CRPS is a proper scoring rule, often used in forecasting scenarios, that compares the full predicted distribution with the observed outcome, thus it also measure calibration, and is defined as:

$$\text{CRPS}(\hat{\boldsymbol{y}}, \boldsymbol{y}, \boldsymbol{\sigma}^2) = \frac{1}{N} \sum_{i=1}^{N} \sum_{k=1}^{3} \text{CRPS}_{\mathcal{N}}(y_i^k; \hat{y}_i^k, \sigma_i^{k\,2}), \tag{27}$$

where $\text{CRPS}_{\mathcal{N}}$ denotes the closed-form CRPS for a univariate Gaussian distribution: To compute the CRPS for a univariate Gaussian predictive distribution with mean $\hat{y}$, variance $\sigma^2$, and ground truth $y$, we use the following closed-form expression:

$$\text{CRPS}_{\mathcal{N}}(y; \hat{y}, \sigma^2) = \sigma \cdot \left( z(2\Phi(z) - 1) + 2\phi(z) - \frac{1}{\sqrt{\pi}} \right), \tag{28}$$

where $z = \frac{y - \hat{y}}{\sigma}$, $\phi(z)$ and $\Phi(z)$ are the standard normal probability density function and cumulative distribution function, respectively.

### B.2. Ammonia and Silica Glass Systems

**Dataset Generation and Availability:** For the ammonia system, we utilize the dataset introduced by Tan et al. (2023), which comprises a small number of $NH_3$ molecular geometries sampled across a range of energy configurations. The dataset is publicly available at: `https://archive.materialscloud.org/records/tz6a0-hsb25`. We follow the same training/validation/test split protocol as Tan et al. (2023).

For the silica glass system, we also use the dataset from Tan et al. (2023), which consists of large $SiO_2$ structures extracted from diverse, long molecular dynamics trajectories. These trajectories span a range of physical conditions, including temperature equilibration, cooling, uniaxial tension, compression, and more. The dataset is publicly available at: `https://archive.materialscloud.org/records/tz6a0-hsb25`. We adopt the same validation/test split protocol as Tan et al. (2023), and vary the size of the training set (32, 128, and 1024 samples).

Additional details regarding the DFT calculations used to generate both datasets are provided in the Supplementary Information of Tan et al. (2023).

**Metrics:** We follow the same experimental procedure as Tan et al. (2023): for prediction we compute the mean absolute error between predicted energy and forces $\hat{E}, \hat{\boldsymbol{F}}$ against true ones:

$$\begin{aligned}
\text{MAE}(\hat{E}, E) &= \frac{1}{N} \sum_{i=1}^{N} \sum_{k=1}^{3} |E_i - \hat{E}_i|, \\
\text{MAE}(\hat{\boldsymbol{F}}, \boldsymbol{F}) &= \frac{1}{3N} \sum_{i=1}^{N} \sum_{k=1}^{3} |F_i^k - \hat{F}_i^k|.
\end{aligned} \tag{29}$$

where the index $k$ denotes the spatial dimensions and $i$ indexes the data points.

To evaluate uncertainty, we compute the Expected Calibration Error on forces, following the approach proposed by (Tan et al., 2023). This choice is motivated by the fact that forces are more sensitive to overfitting and distributional shift, and thus provide a more informative measure of epistemic uncertainty than energy-based metrics. We estimate uncertainties using the mean force per structure, denoted by $\bar{F}_i$ for structure $i$, which is obtained by averaging the per-atom forces and flattening across the spatial dimensions. The corresponding standard deviation, $\bar{\sigma}_i$, is computed as the square root of the average variance norm per atom, following the procedure in (Tan et al., 2023). The ECE is then computed as:

$$\text{ECE} = \int_0^1 |p - p_{obs}| \, dp \quad , p_{obs} = \frac{1}{N} \sum_{i=1}^{N} \mathbb{I} \left\{ \bar{F}_i \leq \mathbb{Q}_p[\hat{\bar{F}}_i, \bar{\sigma}_i] \right\} \tag{30}$$

*Table 5.* Summary of main training-related hyperparameters for different experiments. These hyperparameters are shared among different models. Adam is the optimiser in Kingma and Ba (2014), while AdamW is the one in Loshchilov and Hutter (2017).

| Hyper-parameter | NBody | Ammonia | Silica | MATPES |
|---|---|---|---|---|
| Optimization Epochs | 10000 | 500 | 50 | 25 (5 AL steps) |
| Batch Size | 100 | 64 | 8 | 128 |
| Optimiser | Adam | Adam | AdamW | AdamW |
| Loss | MSE | MAE | HuberLoss(0.01) | HuberLoss(0.01) |
| Maximum learning rate | $5e-4$ | $1e-3$ | $1e-5$ | $1e-5$ |
| Learning rate scheduling | Const. | Reduce On Plateau | Cosine | Const. |
| Gradient Clip Norm Value | NA | NA | 0.5 | 0.5 |
| Warmup Iteration Phase | NA | NA | 5% | N |

where $\mathbb{Q}$ is the gaussian quantile function. The integral is computed by the trapezoidal rule. We further evaluate the rank correlation between predicted variance $\bar{\sigma}_i^2$ and squared error norm per atom using the Spearman correlation coefficients using the Scipy (Virtanen et al., 2020) implementation.

### B.3. MATPES Dataset

**Dataset Availability:** We use the PBE version of the MATPES dataset (Kaplan et al., 2025), which is publicly available at `https://matpes.ai/` . For the zero active learning (AL) step, the dataset is split as follows (fractions of the full dataset): initial training set $= 0.001$, pool set $= 0.025$, test set $= 0.79$, and validation set $= 0.124$. At each AL step, $10\%$ of candidate structures are selected from the pool loader. Selection is either random or based on the top $10\%$ most uncertain structures, where uncertainty is quantified as the standard deviation of the predicted energies.

**Metrics:** We follow the same metrics as in the Silica experiment. Prediction accuracy is evaluated using the mean absolute error for energies and forces:

$$\text{MAE}(\hat{E}, E) = \frac{1}{N} \sum_{i=1}^{N} \sum_{k=1}^{3} |E_i - \hat{E}_i|,$$

$$\text{MAE}(\hat{\boldsymbol{F}}, \boldsymbol{F}) = \frac{1}{3N} \sum_{i=1}^{N} \sum_{k=1}^{3} |F_i^k - \hat{F}_i^k|. \tag{31}$$

where $i$ indexes the data points and $k$ indexes the spatial dimensions. For uncertainty quantification, we use the standard deviation of the predicted energies across stochastic samples.

$$\sigma_E = \frac{1}{N-1} \sqrt{\sum_{i=1}^{N} \left(E_i - \hat{E}_i\right)^2}. \tag{32}$$

## C. Architectures and Training Details

All our models have been trained on a single 64GB A-100 Nvidia GPU. We report in Table 5 the hyperparameters used across experiments shared among different models. All models are trained on the training dataset and evaluated on the test dataset. The validation dataset is used to select the best checkpoint, corresponding to the smallest validation loss. The validation loss is computed using the same loss function as in training, but evaluated on the validation set.

### C.1. NBody System

We benchmark two architectures for this task: a standard Graph Neural Network (GNN) and the Equivariant Graph Neural Network (EGNN), as described in the main text. These two architectures are also used for the probabilistic baselines.

For the GNN baseline, each particle's initial position and velocity are embedded through a linear layer to produce the initial node features $\mathbf{h}^0$, which are passed to the first GNN layer. Edge attributes are derived from pairwise charge interactions, computed as $a_{ij} = c_i c_j$, and are likewise passed through a linear layer before being used in message passing. Both the

message and update functions are implemented as a four-layer MLP of the form

$$64 \rightarrow \text{swish} \rightarrow 64 \rightarrow \text{swish} \rightarrow 64 \rightarrow \text{swish} \rightarrow 64.$$

To compute the final particle positions, we use a regression head consisting of a two-layer MLP:

$$64 \rightarrow \text{swish} \rightarrow 64 \rightarrow \text{swish} \rightarrow 3.$$

For the EGNN baseline, we employ the velocity variant from Satorras et al. (2021), where the initial velocity norm of each particle is mapped through a linear layer and included in the initial node features $\mathbf{h}_i^0$. Particle charges are again encoded as edge attributes via the pairwise product $a_{ij} = c_i c_j$. The message and update functions have the same parameterisation as in the GNN model. Following Satorras et al. (2021), no regression head is used, as the network directly predicts particle positions.

Regarding the baseline methods, for MC Dropout we follow Gal and Ghahramani (2016) and apply dropout with varying probabilities after all linear layers of the message-passing network. For Deep Ensemble, we train the same architecture four times with different random initialisation seeds.

Finally, for BLIP, we employ a compact invariant posterior network consisting of two linear layers of size $64$ with swish activations, used for both the message and update coefficients. Dropout probabilities $p$ are predicted and converted to coefficients via $p/(1-p)$. The message inference network head takes as input the edge attribute $a_{ij}$ together with $\mathbf{h}_i^0$ and $\mathbf{h}_j^0$, whereas the update inference network head only takes $\mathbf{h}^0$. For stability, we clamp the maximum VAD coefficients to $4.0$. During training, we use a prior dropout probability of $0.5$ and weight the regularisation term $D_{\mathrm{KL}}$ by $\lambda = 0.01$.

### C.2. Ammonia Molecule System

We adopt the PaiNN architecture (Schütt et al., 2021) as our machine learning interatomic potential. To ensure a fair comparison, we use the same hyperparameter settings as in Tan et al. (2023), which are publicly available at: `https://github.com/learningmatter-mit/UQ_singleNN`.

For BLIP, we employ a compact invariant posterior network consisting of two linear layers of size $64$ with swish activations, used for both the message and update coefficients:

$$64 \rightarrow \text{swish} \rightarrow 64 \rightarrow \text{swish} \rightarrow \texttt{head-dim}.$$

The message inference network head takes as input the squared interatomic distance $\|\boldsymbol{r}_i - \boldsymbol{r}_j\|^2$, whereas the update inference network head takes only an embedding of the element atomic number $Z$. For stability, we clamp the VAD coefficients to $4.0$. During training, we use a prior dropout probability of $0.5$ and weight the regularisation term $D_{\mathrm{KL}}$ by $\lambda = 10$.

### C.3. Silica Glass System

We adopt the ORB v3 architecture (Rhodes et al., 2025) as our machine learning interatomic potential, pretrained on ab initio molecular dynamics data from OMAT 24 (Barroso-Luque et al., 2024). The Deep Ensemble variant finetunes four different models with different initialisation seeds.

For BLIP, we employ a compact posterior network composed of four linear layers with hidden size $64$ and swish activations for both the message and update functions:

$$64 \rightarrow \text{swish} \rightarrow 64 \rightarrow \text{swish} \rightarrow 64 \rightarrow \text{swish} \rightarrow 64 \rightarrow \text{swish} \rightarrow \texttt{head-dim}.$$

The edge network takes as input a concatenation of the pairwise distance vector (normalised to unit length) and 20 Gaussian radial basis functions applied to the raw edge distance. The node network processes atomic number embeddings. For stability, we clamp the VAD coefficients to the range $[10^{-5}, 0.11]$. During training, we use a prior dropout probability of $0.01$ and weight the regularisation term $D_{\mathrm{KL}}$ by $\lambda = 0.001$. The lower clamp values, $\lambda$, and reduced prior dropout are chosen because we find that the original ORB model weights are highly sensitive to even minimal perturbations, likely because they were not trained with dropout or other weight-perturbation techniques. In addition, we use a different learning rate ($1e-4$) for the inference network to enable faster adaptation compared to the pretrained weights.

### C.4. MATPES dataset

We employ the same hyperparameters as for the Silica Glass System.

## D. Algorithms

This section reports the pseudocode algorithms for training and performing inference using a BLIP.

**Training with BLIPs:**  Algorithm 2 outlines the training procedure for Bayesian Learned Interatomic Potentials. Given a graph representing an atomic configuration with initial node features $\boldsymbol{h}^0$ and edge attributes $a_{ij}$, the inference network $E_\psi$ first produces variational adaptive dropout (VAD) coefficients $\boldsymbol{\alpha}, \boldsymbol{\beta}$ for each Bayesian linear layer. These parameters define the uncertainty-aware weight distributions used in the subsequent message passing steps. At each layer $l = 0, \ldots, L-1$, the node features $\boldsymbol{h}^l$ are updated using a message passing function that incorporates stochastic sampling via the local reparameterization trick (see Algorithm 1). After the final layer, the atomic representations $\boldsymbol{h}^{1:L}$ are aggregated to predict the total energy $E$. If the potential field is conservative, forces $\mathbf{F}$ are obtained by differentiating the energy; otherwise predicted directly from the final layer features $\boldsymbol{h}^L$. The loss combines mean absolute errors on energy and forces with the KL divergence regularizer.

---

**Algorithm 2 Training Bayesian Learned Interatomic Potentials**: Given a graph representing an atomic configuration with initial node features $\boldsymbol{h}^0$ and edge attributes $a_{ij}$, the model is trained by sampling Bayesian weights, performing stochastic message passing, and computing energies and forces. The loss combines mean absolute errors on energy and force predictions with a KL divergence regularizer depending only on the VAD coefficients.

---

1: **while** not converged **do**
2:     $\boldsymbol{\alpha}, \boldsymbol{\beta} \leftarrow E_\psi(\boldsymbol{h}^0, a_{ij})$ {Encode inputs to obtain VAD coefficients for all layers}
3:     **for** layer $l = 0, \ldots, L-1$ **do**
4:         $\boldsymbol{h}^{l+1} \leftarrow \text{MessagePassing}(\boldsymbol{h}^l, a_{ij}, \boldsymbol{\alpha}^l, \boldsymbol{\beta}^l)$ {Stochastic message passing using Alg. 1}
5:     **end for**
6:     Compute energy $E$ from $\boldsymbol{h}^{1:L}$ by pooling
7:     **if** field is conservative **then**
8:         $\mathbf{F} \leftarrow -\nabla E$
9:     **else**
10:        Compute forces $\mathbf{F}$ directly from $\boldsymbol{h}^L$
11:     **end if**
12:     $\mathcal{L} \leftarrow \lambda_\text{E}\text{MAE}_\text{E} + \lambda_\text{F}\text{MAE}_\text{F} + \lambda D_\text{KL}(\boldsymbol{\alpha}, \boldsymbol{\beta})$ {Loss includes MAEs and KL divergence term from equation 11}
13: **end while**

---

**Inference with BLIPs.**  Uncertainty quantification is performed via multiple stochastic forward passes through the model using Monte Carlo (MC) sampling. Importantly, the inference network is evaluated only once to extract the variational parameters $\boldsymbol{\alpha}, \boldsymbol{\beta}$, which are reused across MC samples. This amortization significantly reduces computational overhead. While more accurate moment-matching approximations could be used for the output distributions, we leave this for future work. Finally, if only predictive means are needed and uncertainty estimates are not required, a single deterministic forward pass can be performed using the mean of the variational distributions (i.e., without stochastic sampling), effectively matching the performance speed of a standard MLIP.

## E. Computational Cost Analysis

A central motivation of our approach is to obtain high-quality uncertainty estimates and mean predictions with minimal additional computational overhead. In this section, we quantify the training, inference, sampling and memory cost of our method and compare it against deep ensembles. In Table 6 we summarise relevant statistics as model parameters, training and inference FLOPs.

**Training Cost:**  Our method trains a single backbone model jointly with a lightweight inference network $E$ that contains only $p\%$ of the parameters of the main model (typically $0.1\%$ to $5\%$). Let $C_\text{main}$ denote the cost in FLOPs of training the

---

**Algorithm 3 BLIP Inference with Uncertainty Quantification**

---

1: $\boldsymbol{\alpha}, \boldsymbol{\beta} \leftarrow E_{\boldsymbol{\psi}}(\boldsymbol{h}^0, a_{ij})$ {Encode inputs to obtain VAD coefficients once}
2: energies $\leftarrow [\,]$, forces $\leftarrow [\,]$
3: **for** each MC sample $s = 1, \ldots, S$ **do**
4:    $\boldsymbol{h}^0 \leftarrow$ initial node features
5:    **for** layer $l = 0, \ldots, L-1$ **do**
6:       $\boldsymbol{h}^{l+1} \leftarrow \text{MessagePassing}(\boldsymbol{h}^l, a_{ij}, \boldsymbol{\alpha}^l, \boldsymbol{\beta}^l)$ {Sample weights via local reparametrization (Alg. 1)}
7:    **end for**
8:    Compute energy $E^{(s)}$ from $\boldsymbol{h}^{1:L}$ by pooling
9:    **if** field is conservative **then**
10:       $\mathbf{F}^{(s)} \leftarrow -\nabla E^{(s)}$
11:    **else**
12:       Compute $\mathbf{F}^{(s)}$ directly from $\boldsymbol{h}^L$
13:    **end if**
14:    Append $E^{(s)}$ to energies, $\mathbf{F}^{(s)}$ to forces
15: **end for**
16: Compute mean and variance statistics over energies and forces

---

**Algorithm 4 BLIP Inference with only mean Energy and Forces prediction**

---

1: **for** layer $l = 0, \ldots, L-1$ **do**
2:    $\boldsymbol{h}^{l+1} \leftarrow \text{MessagePassing}(\boldsymbol{h}^l, a_{ij}, \mathbf{0}, \mathbf{0})$ {Use mean weights, no perturbation}
3: **end for**
4: Compute energy $E$ from $\boldsymbol{h}^{1:L}$ by pooling
5: **if** field is conservative **then**
6:    $\mathbf{F} \leftarrow -\nabla E$
7: **else**
8:    Compute $\mathbf{F}$ directly from $\boldsymbol{h}^L$
9: **end if**

---

base model, and $C_{\text{inf}}$ the one for the inference network. The total training cost of BLIP is therefore

$$C_{\text{BLIP}} = C_{\text{main}} + C_{\text{inf}} \approx (2 + \epsilon)\, C_{\text{main}},$$

where $\epsilon = C_{\text{inf}}/C_{\text{main}} \approx p/100 \ll 1$. The factor of two comes from the local reparameterization trick, which requires computing both the mean and variance. In contrast, an ensemble of $N$ independently trained models incurs a cost of

$$C_{\text{ens}} = N\, C_{\text{main}},$$

which is $N$-times larger than training a single model. For commonly used ensemble sizes ($N = 5$), this results in roughly a $5\times$ increase in both computation and memory footprint.

*Table 6.* FLOPs breakdown for ORBv3-inf-omat24 (Base) model for training and inference.

| | | Base | Base + BLIP | N× Ensemble Base |
|---|---|---|---|---|
| | Params | 21.02 | 21.05 | $21.02 \times N$ |
| | Params increase from Base | $\times 1$ | $\times 1.001$ | $\times N$ |
| *Training* | FLOPs / atom ($\times 10^{12}$) | 1.65 | 3.1 | $1.65 \times N$ |
| | FLOPs increase from Base | $\times 1$ | $\times 1.9$ | $\times N$ |
| *Inference* | FLOPs / atom ($\times 10^{12}$) | 1.65 | 1.65 | $1.65 \times N$ |
| | FLOPs increase from Base | $\times 1$ | $\times 1$ | $\times N$ |

**Inference Cost:** At inference time, our method evaluates a single forward pass through the backbone as usual, since we set the variational dropout coefficient to zero. **This approach does not add any overhead compared to the cost of evaluating the backbone model**. On the other hand, an ensemble of $N$ models requires $N$ separate forward passes, thus scaling linearly in size.

**Sampling Cost:** For performing a single sample from the posterior, the forward FLOPs required are the same as during training. Advanced techniques such as vectorial-mapping, moment matching or moment linearization could be employed to drastically reduce the computational cost of predictive distribution moment estimation, but we leave this for future work.

