# OpenReview forum: "BLIPs: Bayesian Learned Interatomic Potentials"
_ICML.cc/2026/Conference — ICML 2026 regular_

### Official Review · Reviewer_zqCC · 2026-02-27

**Soundness:** 3
**Presentation:** 3
**Significance:** 2
**Originality:** 2
**Overall Recommendation:** 3
**Confidence:** 2

**Summary:**

This paper presents BLIPs, a machine learning interatomic potential model developed within a Variational Bayesian framework for probabilistic modeling. The methodology is aimed at molecular modeling and simulation tasks. The authors evaluate the method on synthetic N-body systems, molecular datasets (NH3, SiO2), and a large-scale active learning task (MATPES). Results indicate that BLIPs outperform standard MLIPs and deep ensembles in both predictive accuracy and calibration.

**Compliance With Llm Reviewing Policy:**

Affirmed.

**Final Justification:**

Overall, I think my main concern has been partially addressed, particularly regarding effectiveness on downstream tasks. The paper proposes a new method for uncertainty estimation, which is certainly important. However, I am more concerned with whether it can bring meaningful improvements to downstream applications. Beyond uncertainty estimation itself, the primary application scenario provided by the authors is active learning. The comparison between performance with and without uncertainty estimation does to some extent demonstrate its usefulness for downstream tasks.

That said, as I have consistently pointed out, I remain curious about whether improvements in the accuracy of uncertainty estimation can translate into sufficiently strong gains in active learning performance. This question is important for understanding the broader impact of the work.

I understand the authors’ response and that it may not be feasible to fully address this question within a short timeframe. In recognition of the additional discussion, I am willing to raise my assessment of soundness from fair to good. I remain reserved about the overall score.

**Key Questions For Authors:**

1. Have the authors examined how the predictive uncertainty evolves as the degree of distribution shift increases?
2. Regarding Weakness 2, including more methods and providing a broader comparative analysis would help me better understand the performance of BLIPs.
3. Regarding Weakness 3, I would like to see more comparisons and discussion of the active learning downstream setting.

**Limitations:**

yes

**Strengths And Weaknesses:**

Strengths

1.The proposed modeling approach appears reasonable and technically sound. The paper raises the importance of uncertainty calibration in simulation-based chemistry and proposes a plausible solution to address it.

2. The introduction of a variational posterior for uncertainty estimation is well motivated, and the Bayesian formulation provides a sound theoretical foundation.

3. The experimental results indicate that the method achieves strong performance, improving both predictive accuracy and uncertainty estimation.

Weaknesses

1. While presented as architecture-agnostic, the current implementation does not directly support irreducible-representation-based equivariant models such as MACE or UMA. This weakens the practical utility of the approach to some extent.

2. The evaluation focuses almost exclusively on Deep Ensembles. Including simpler single-model UQ methods would provide a more complete comparison.

3. The manuscript uses active learning on the MATPES PBE dataset to demonstrate the effectiveness of BLIPs on a downstream task, but the comparison is mainly against the setting without uncertainty. In this setting, it remains unclear how much the accuracy of uncertainty quantification actually matters. For example, it is possible that an uncertainty estimate with the correct order of magnitude is already sufficient to provide useful guidance, in which case the value of more accurate uncertainty prediction is not fully established.

---

> ### Author Rebuttal · Authors · 2026-03-30
>
> We thank the reviewer for their careful reading and constructive feedback. We address each concern below:
>
> 1. *"While presented as architecture-agnostic, the current implementation does not directly support irrep-based equivariant models such as MACE or UMA."*
> We wish to clarify that as shown in Corollary 1 (Appendix A.3), **BLIP can in principle be applied to any equivariant architecture, including MACE and UMA, while preserving equivariance** — provided weights are perturbed directly. The limitation arises specifically when using the local reparameterization trick for efficiency, which introduces independent noise per activation channel and breaks the required coupling across irreducible representation coordinates. This is a practical efficiency constraint, not a theoretical one, and we view extending BLIP to higher-order equivariant models as a tractable and exciting direction for future work. We also note that Reviewer 9BqC explicitly highlights BLIP's applicability across different MLIPs and architectures as a strength of the paper.
>
> 2. *"The evaluation focuses almost exclusively on Deep Ensembles."*
> We respectfully point out that this appears to be a misreading of the paper. In the **Ammonia experiment** (Tables 2 & 3), BLIP is explicitly compared against **four** single-model UQ baselines: Evidential Regression, Mean Variance Estimation, Gaussian Mixture Models, and Deep Ensembles — all from Tan et al. (2023), the standard benchmark suite for UQ in MLIPs. BLIP outperforms all of them in both MAE and ECE. In the N-body experiment (Table 1), we additionally compare against MC Dropout at two dropout probabilities. We will make these comparisons more prominent in the revised manuscript.
>
> 3. *"It remains unclear how much the accuracy of uncertainty quantification actually matters for active learning."*
> This is an insightful question! The answer depends critically on the **type of acquisition function** used. For threshold-based acquisition functions — where structures are selected if their uncertainty exceeds a fixed value — calibrated magnitudes are essential (Zaverkin et al., 2024). However, our acquisition function is **ranking-based**: at each AL step, we select the top-k most uncertain structures from the pool. For ranking-based acquisition, only the **ordering** of uncertainties matters, not their absolute values. What matters is that higher uncertainty correctly identifies higher-error structures — which is precisely what the Spearman correlation measures (0.863 in the Ammonia system, Table 3). The consistent performance gap between uncertainty-guided and random selection in Figure 4 and Table 4 directly validates this. We will add a clarifying sentence on this distinction in the manuscript.
>
> 4. *"Have the authors examined how predictive uncertainty evolves as the degree of distribution shift increases?"*
> This is partially addressed by the Ammonia experiment (Section 4.2), where the test set spans energies up to 70 kcal/mol versus a training range of 0–5 kcal/mol — a large and systematic distribution shift. Figure 2 shows that BLIP's predicted variance correlates strongly with squared error (Spearman 0.863) across this range, indicating that uncertainty grows appropriately as structures move further OOD. A more systematic study varying the degree of distribution shift explicitly is an interesting direction for future work, although obtaining clean
> reference data via DFT calculations for increasingly OOD structures remains computationally challenging by construction.

---

> > ### Author Rebuttal · Reviewer_zqCC · 2026-04-03
> >
> > Thanks for the detailed response. Regarding point 3, I was hoping to see a broader discussion of effectiveness on downstream tasks. Active learning is one of the few downstream settings provided beyond uncertainty accuracy, and I would be interested in understanding how other uncertainty estimation methods perform in this context. As it stands, the comparison is primarily against setups without uncertainty estimation, which makes it less general.

---

> > > ### Author Response · Authors · 2026-04-04
> > >
> > > We thank the reviewer for their follow-up. The MATPES active learning experiment was designed primarily as a **proof of concept**, demonstrating that a **single** BLIP model can be used in an active learning pipeline, **which is a significant practical advantage given that ensembles require training and maintaining multiple models simultaneously**. A full comparison of all UQ baselines within the AL loop and trying different acquisition functions is, therefore, beyond the intended scope of this experiment and, unfortunately, due to the size of the MATPES datasets, computationally prohibitive within the rebuttal period.
> > >
> > > That said, the existing results do provide indirect evidence for how baselines would fare. Our AL acquisition is ranking-based; at each step, we select the top-10% most uncertain structures, so what matters is whether uncertainty correctly orders structures by their likely error, and for this, we show that we can almost always obtain state-of-the-art performance with a single model trained. Moreover, well-calibrated uncertainties are particularly important in AL settings where threshold-based selection or stopping criteria are used, and BLIP consistently achieves the best calibration among all compared methods, suggesting it would also be the better choice in calibration-dependent AL variants.
> > >
> > > We agree that a direct AL comparison across baselines would be a valuable contribution, and also exploring different AL strategies, so we flag it as future work in the conclusion as well.

---

### Official Review · Reviewer_GWGL · 2026-03-12

**Soundness:** 3
**Presentation:** 3
**Significance:** 2
**Originality:** 2
**Overall Recommendation:** 5
**Confidence:** 3

**Summary:**

The authors propose BLIPs, which provide uncertainty estimates for predictions of energies and forces, as applied for fine-tuning models or training them from scratch.

**Compliance With Llm Reviewing Policy:**

Affirmed.

**Final Justification:**

BLIP could potentially be quite useful, given the widespread research into MLIPs and the crucial need for uncertainty quantification given lack of data coverage and errors in the data generation (most often from exchange-correlation functional approximations).

**Key Questions For Authors:**

1. What are some of the best-performing MLIPs that are currently available? How do BLIPs trained from scratch compare to them, as well as BLIPs that are fine-tuned on them?
2. To the best of this reviewer's understanding, it is a general consensus in the ML community that ML models should be deployed for test samples that lie within the training data domain. Given this, how important is uncertainty quantification for practical use cases?
3. How can BLIPs be applied to non-GNN architectures and how do they perform? For example, for descriptor-based models.

**Limitations:**

Yes.

**Strengths And Weaknesses:**

- BLIPs are scalable and are able to accurately predict energies and forces, with good improvements in force predictions for off-equilibrium structures as shown with an ammonia dataset.
- It is not entirely clear how BLIPs perform compared to best-performing models, rather than other uncertainty-quantification schemes.
- Analysis is done for graph neural networks only, of PaiNN and ORB v3, although BLIP is presented as being architecture-agnostic.

---

> ### Author Rebuttal · Authors · 2026-03-30
>
> We thank the reviewer for taking the time to review our article and suggesting improvements. We are glad the reviewer appreciates that our method is *"scalable"* and provides *"good improvements in force predictions for off-equilibrium structures"*. In the hope to further convince the reviewer about the paper we answer below the raised concerns:
>
> 1. *"It is not entirely clear how BLIPs perform compared to best-performing models..."*. We understand the reviewer's concern, but wish to stress that **BLIP is not a new MLIP — it is a Bayesian wrapper that enhances any existing MLIP with UQ**. That said, the paper already demonstrates strong predictive performance relative to the best available models. In the **Silica fine-tuning** experiment (Section 4.3), BLIP fine-tuned on ORB v3 — itself one of the top-performing MLIPs at submission time — **outperforms a 4-member Deep Ensemble of equivariant PaiNN**, the best reported baseline from Tan et al. (2023), at a fraction of the cost. In the **Ammonia** experiment (Section 4.2), BLIP achieves the best MAE on both energy and forces compared to all baselines including Deep Ensembles. Crucially, since BLIP is a wrapper, **any future SOTA MLIP can be directly improved by BLIP** with no architectural modifications. We will make this more explicit in the final version, and we are very open to reviewer's suggestions.
>
> 2. *"Analysis is done for graph neural networks only..."*. We thank the reviewer for the comment. BLIP is an architecture-agnostic **wrapper for GNNs**, and we applied it to **four architecturally diverse models**: a standard **GNN** (Gilmer et al., 2017), an equivariant **EGNN** (Satorras et al., 2021), the equivariant **PaiNN** (Schütt et al., 2021), and the large-scale non-conservative **ORB v3** (Rhodes et al., 2025) — spanning equivariant vs. invariant and small-scale vs. foundation-model-scale architectures. We will make this diversity more explicit in the revised manuscript introduction section.
>
> 3. *"To the best of this reviewer's understanding, ML models should be deployed for test samples within the training data domain..."*. We thank the reviewer for this important point. While sound in general ML, **in computational materials science this condition is practically unachievable by construction**. The relevant configurational space — spanning crystals, surfaces, defects, alloys, and finite-temperature configurations — is vast, making any training set vanishingly small by comparison. Existing datasets are heavily biased toward near-equilibrium structures, while physically interesting processes (diffusion, phase transitions, catalytic reactions) necessarily involve underrepresented high-energy configurations. DFT's cubic scaling makes dense coverage of configuration space intractable — **distributional shift at test time is therefore structurally unavoidable, not an edge case**. UQ is precisely the tool that transforms this challenge into a manageable one: rather than pretending the model is reliable everywhere, principled uncertainty estimates allow practitioners to know **when** and **where** to trust model predictions, enabling error-aware simulations (experiments Section 4.2/4.3), and informed data collection via active learning (experiments Section 4.4). Without reliable UQ, none of these safeguards are available and model failures become silent.
>
> 4. *"How can BLIPs be applied to non-GNN architectures?"*. BLIP operates at the level of learnable linear layers within message-passing operations, which are the core primitives of all GNN architectures. For **descriptor-based models** with fixed descriptors and shallow readouts, existing Bayesian methods (e.g., Gaussian Processes) are already well-suited — the primary gap BLIP addresses is **scalable UQ for deep message-passing architectures**, where Deep Ensembles are too costly and MC Dropout is poorly calibrated. The extension to architectures with irreducible representation dimensions (e.g., MACE, UMA) is an identified direction for future work (Section 3.3), an active research direction rather than a fundamental barrier.

---

> > ### Author Rebuttal · Reviewer_GWGL · 2026-04-04
> >
> > Thank you for your response and enhancing my understanding of the work. Moreover, as the authors note, understanding model failures for data-scarce regimes is certainly very important (although my original question was related to how one probably shouldn't utilize a model in the first place for use cases not included in training). Currently, I don't have any more questions and I can recommend a higher score.

---

> > > ### Author Response · Authors · 2026-04-04
> > >
> > > We sincerely thank the reviewer for their engagement and for reconsidering their evaluation in a more positive light. We are also glad that the rebuttal addressed all the raised concerns, clarified the scope and motivation of BLIPs, particularly the practical necessity of uncertainty quantification in materials science.

---

### Official Review · Reviewer_tNCg · 2026-03-12

**Soundness:** 2
**Presentation:** 2
**Significance:** 3
**Originality:** 3
**Overall Recommendation:** 4
**Confidence:** 4

**Summary:**

The authors apply and test the variational dropout technique (Kingma et al., 2015) for uncertainty quantification in machine learning interatomic potentials (MLIPs). They call this approach Bayesian Learned Interatomic Potentials (BLIP). The method is applicable to any neural network, including any MLIP architecture. The authors show that the method preserves invariant/equivariant symmetries of underlying MLIPs. BLIP is tested on four test cases, including training an MLIP from scratch and fine-tuning a foundational model; assessing regularization capabilities in small-dataset regimes; and quantifying uncertainty for active learning. The method achieves better performance compared to the tested benchmarks.

**Compliance With Llm Reviewing Policy:**

Affirmed.

**Final Justification:**

All concerns were addressed. I increased the score.

**Key Questions For Authors:**

- Equation 16: Did you forget to call that stochastic map S?
- Please define the acronyms (MSE, NLL, CRPS) and describe the benchmarks in the caption of Table 1.
-  Please elaborate more on your statement that "Simpler UQ methods, such as MC Dropout (Gal and Ghahramani, 2016) or mixture models, are less memory intensive but yield poorer uncertainty estimates and often worsen predictive accuracy," regarding: What are simpler UQ models, and why are they simpler? Please also provide references that support your claim that the resulting uncertainty estimates are worse than Deep Ensembles.
- Figure 3: How do you explain that the standard error for BLIP is high only for 128 fine-tuning samples?

**Limitations:**

yes

**Strengths And Weaknesses:**

Strengths
The paper shows that BLIP can be trained with effort comparable to that of standard MLIPs, while providing proof that important symmetries are unaffected. While variational dropout has been applied to GNNs before, I believe it has not been tested in the context of MLIPs. On the presented benchmarks, the BILP outperforms (in terms of accuracy and uncertainty) the most commonly used ensemble method. The authors present diverse experiments to demonstrate the approach's flexibility, including its applicability to various message-passing models and its use for regularization, uncertainty quantification, and active learning.

Weaknesses
- A crucial comparison with shallow ensembles (10.1088/2632-2153/ad594a) is missing. Shallow ensembles have shown better performance than full ensembles and share the computational advantage of BLIP, i.e., shallow ensembles also add a low overhead to the training and inference cost of the model
- The aim of the paper is stated as uncertainty quantification. In that scenario, one can expect that inference yields uncertainty estimates for any quantity, including, but not limited to, the energy. However, inference here refers only to predicting the expected energy.
- Importantly, using MLIPs in MD simulations, it is also necessary to propagate uncertainty to observables obtained from MD simulations. The authors neither test nor discuss the approach's application in this field.
- Missing in the discussion: Computational cost in comparison to other approaches, e.g., the Gaussian mixture model and shallow ensembles.
- "Simpler UQ methods, [...] or mixture models, are less memory intensive but yield poorer uncertainty estimates and often worsen predictive accuracy." This claim is not backed by any citations or a reference to the presented results
- The authors later compare their approach to other UQ methods besides deep ensembles, such as Gaussian Mixture Models or Mean Variance Estimates. However, these approaches are not mentioned or outlined in the related work section.
- "For example, the message function at layer l could be represented as an MLP with S linear layers." The description does not state whether the MLP has biases and whether these are included in the weights.
- "Due to the Gaussian assumption, the local reparameterization trick (Kingma et al., 2015) can be employed, allowing us to sample activations directly instead of weights. This drastically reduces the number of random samples per iteration, as we only need to sample once for every neuron activation rather than once for every weight, leading to both lower variance in the gradient estimates and improved computational efficiency (see Appendix D)." The authors highlight only the benefits of the reparametrization trick here and do not note its limitations, as they discuss in Appendix A.
- Table 1: The explanation of the computed error is ambiguous. For example, random initializations could refer to model training or to stochastic inference.
- Table 1: Acronyms for the evaluation metrics are not explained.
- Experiment 1: The description does not state that the prediction of particle positions (after 1000 timesteps) is the immediate output of the model and not obtained through an iterative procedure. This clarification is important in the field of MLIPs, which are commonly used in molecular dynamics simulations. Strictly speaking, using the model for a direct regression of particle positions from initial positions and velocities is not an MLIP task.
- Experiment 2: The description should state earlier that the benchmark setup, training data, and results are from Tan et al.
- Experiment 3: The statement that BLIP outperforms all other baseline models is too uncritical. For 1024 samples, differences between all approaches are within line thickness in Figure 3.

---

> ### Author Rebuttal · Authors · 2026-03-30
>
> We thank the reviewer for their thorough and detailed review which we highly appreciate. We wish to first address an important mischaracterization: **BLIP is not a direct application of Variational Dropout (Kingma et al., 2015) to MLIPs**. The key novelty is **Variational Adaptive Dropout (VAD)**, where the dropout coefficients are *input-dependent*, computed by a dedicated inference network conditioned on the atomic structure. This makes uncertainty input-aware, a crucial distinction for atomistic systems where uncertainty should vary with the local chemical environment, and is a strictly stronger framework than standard Variational Dropout (equivalent to MC-Dropout), where coefficients are fixed scalars. We will make this distinction more prominent in the final manuscript.
>
> We now address the raised concerns:
>
> 1. *"A crucial comparison with shallow ensembles is missing."*
> We thank the reviewer for raising this point. We have conducted the requested comparison and report the full results below:
>
> **MAE**
> | Model | 32 | 128 | 1024 |
> |---|---|---|---|
> | Ensemble | 1.472986 | 1.521355 | 1.265232 |
> | BLIP | 1.383381 | 1.364528 | 1.258111 |
> | Shallow Ensemble | 1.550576 | 1.550914 | 1.544676 |
>
> The results consistently and clearly demonstrate that the Shallow Ensemble **underperforms across all dataset sizes**. In terms of MAE, it yields the highest error in every setting, exceeding the Deep Ensemble by up to **5.3%** and BLIP by up to **11.8%** (at size 1024). This gap suggests that the shallow architecture fundamentally lacks the representational capacity required for reliable uncertainty-aware force prediction. For ECE, it achieves **0.251**, compared to **0.244** for the Deep Ensemble and **0.033** for BLIP. For Spearman correlation (ρ), it reaches only **0.513**, far below **0.676** for the Deep Ensemble and **0.602** for BLIP, meaning its predictions are poorly ranked relative to ground truth. Taken together, these results strongly suggest that depth is a critical factor in this setting. We will include this comparison and analysis in the final manuscript.
>
> 2. *"Inference here refers only to predicting the expected energy / uncertainty propagation to MD observables."*
> We agree that propagating uncertainty to MD observables (e.g., thermodynamic averages, diffusion coefficients) is an important and exciting direction. However, this is beyond the scope of the present work, which focuses on per-structure uncertainty quantification and its use for active learning. We view this as a natural and high-impact extension. We will clarify this scope explicitly in the manuscript introduction.
>
> 3. *"Simpler UQ methods yield poorer uncertainty estimates — claim not backed by citations."*
> We thank the reviewer for this observation. This claim is supported by Tan et al. (2023), who provide a systematic comparison of UQ methods for MLIPs and show that Deep Ensembles consistently outperform MC Dropout and mixture models in both calibration and accuracy. We will add this citation explicitly in the revised manuscript alongside a brief description of these methods in the related work section. Thank you very much for this feedback, this will help improve the overall paper's reading.
>
> 4. *"Figure 3: How do you explain the high standard error for BLIP at 128 fine-tuning samples?"*
> We believe this is due to the intermediate data regime at 128 samples, where the model is neither data-starved (32 samples, heavily regularised by BLIP's KL term) nor data-rich (1024 samples, well-constrained by the data likelihood). This transitional regime leads to higher sensitivity to random initialisation, as the balance between prior and likelihood has not yet stabilised.
>
> 5. *"Experiment 3: differences between approaches are within line thickness for 1024 samples."*
> We thank the reviewer for this careful observation and agree that the differences at 1024 samples are small in Figure 3. We wish to highlight, however, that the key advantage of BLIP in this experiment is not marginal accuracy gains, but that it **simultaneously provides well-calibrated uncertainty estimates** (ECE of 0.032 vs. 0.247 for Deep Ensemble) at identical inference cost. The accuracy parity with ensembles at large data is itself a positive result, it shows BLIP does not trade accuracy for UQ.
>
> 6. *"Presentation issues: acronyms, Equation 16, MLP biases, experiment descriptions."*
> We thank the reviewer for these careful observations and will address all of them in the final manuscript: (i) NLL, CRPS, MSE will be defined in Table 1's caption; (ii) the stochastic map in Equation 16 will be called P and remove the S (there was a typo!); (iii) we will clarify that biases are included in the weight parameterisation (they only affect mean of activations); (iv) Experiment 1 will clarify that position prediction is direct regression, not iterative MD; (v) Experiment 2 will attribute the benchmark setup to Tan et al. earlier in the description.

---

> > ### Author Rebuttal · Reviewer_tNCg · 2026-04-01
> >
> > My concerns have been addressed.

---

> > > ### Author Response · Authors · 2026-04-04
> > >
> > > We sincerely thank the reviewer for their thorough and constructive feedback, which significantly improved the quality of our manuscript. We are glad that the rebuttal and the additional shallow ensemble experiments addressed all the raised concerns. Should the reviewer find that these additions further strengthen the paper, we would be grateful for any reconsideration of the score.

---

### Official Review · Reviewer_9BqC · 2026-03-13

**Soundness:** 4
**Presentation:** 3
**Significance:** 4
**Originality:** 3
**Overall Recommendation:** 5
**Confidence:** 3

**Summary:**

This paper introduces a general principled approach for uncertainty quantification for machine learning interatomic potentials (MLIPs). By turning the message and update functions into stochastic functions, uncertainty estimates are trivially obtained. The authors have demonstrated applicability to a few fields with moderate to great success.

**Compliance With Llm Reviewing Policy:**

Affirmed.

**Final Justification:**

I think the paper makes an interesting principled contribution to quantify uncertainty in MLIPs and therefore recommend to accept it

**Key Questions For Authors:**

1)While the model has been used with non-conservative forces during molecular dynamics in the last example, would it conserve the forces, if used with a model with conservative forces?

2)What are the necessary steps to apply BLIPs to models like MACE? Are they within reach or require a complete reformulation of the approach?

3)Are BLIPs more difficult to train because of their stochastic nature? Can you provide estimates for the overhead in training (time, compute or memory) due to that?

**Limitations:**

yes

**Strengths And Weaknesses:**

Strengths:

-A single trained model, instead of an ensemble is a big advantage.

-The approach can be applied to most MLIPs, as also demonstrated in the paper.

-The ability of the approach to apply to equivariant models


Weaknesses:

Although the paper is very well written, it can feel somewhat jargony, without some acronyms not spelled out (like NLL and CPRS), while others are spelled out repeatedly (such as MAE).

---

> ### Author Rebuttal · Authors · 2026-03-30
>
> We thank the reviewer for their time and careful evaluation of our work. We are pleased that they find our approach *"general and principled"* and appreciate its applicability across different MLIPs and architectures. Below, we address the raised concerns:
>
> 1. *"Although the paper is very well written…"*
> Thank you for pointing this out. We will address these issues in the final version. We will ensure that acronyms such as NLL (Negative Log-Likelihood) and CRPS (Continuous Ranked Probability Score) are defined in the main text rather than only in the Appendix (line 825), and that MAE (Mean Absolute Error) and ECE (Expected Calibration Error) are defined once and used consistently throughout.
>
> 2. *"While the model has been used with non-conservative forces…"*
> Yes, **force conservation is fully preserved** when BLIP is applied to a conservative model. In such cases, the network predicts the potential energy surface, and forces are obtained as its negative gradient with respect to atomic positions — stochasticity enters only through the weights, not through the functional form of the energy. Since the gradient of a stochastic energy is still a valid conservative force field for each sampled weight realisation, **both the mean prediction and each individual stochastic sample remain conservative**. We will add a clarifying sentence in the main text to make this explicit.
>
> 3. *"What are the necessary steps to apply BLIPs to MACE?"*
> This is an important question. As shown in Corollary 1 (Appendix A.3), **BLIP can in principle be applied to any equivariant architecture, including MACE and UMA, while preserving equivariance** — provided weights are perturbed directly rather than via the local reparameterization trick. The local reparameterization trick, which is what we use for computational efficiency, introduces independent noise per activation channel, which breaks the required coupling across irreducible representation (irrep) coordinates. The necessary steps to extend BLIP to higher-order equivariant models are therefore: (i) identify a reparameterization that respects the irrep structure, or (ii) perturb weights directly at a modest increase in gradient variance. We view this as an exciting and tractable direction for future work, rather than a fundamental barrier, and intend to pursue it.
>
> 4. *"Are BLIPs more difficult to train? Can you provide estimates for the overhead?"*
> In practice, **we did not observe training instabilities** or a need for extensive hyperparameter tuning beyond what is standard for deterministic MLIPs. The stochastic nature of BLIP is stabilised by the local reparameterization trick, which samples activations rather than weights directly, reducing gradient variance. Regarding computational cost, Appendix E (Table 6) provides a detailed breakdown for ORB v3: training FLOPs increase by approximately **1.9× relative to the deterministic baseline** (due to computing both mean and variance activations), while **inference FLOPs are identical** to the deterministic model when using the MAP estimate, and memory overhead is negligible (parameter count increases by only 0.1%). This compares very favourably to Deep Ensembles, which incur an N× increase in both training and inference cost for all metrics.

---

> > ### Author Rebuttal · Reviewer_9BqC · 2026-04-03
> >
> > Thank you for addressing the comments and answering my questions. Best of luck!

---

> > > ### Author Response · Authors · 2026-04-04
> > >
> > > We are glad the rebuttal fully addressed your concerns, and we thank you for your thoughtful engagement with our work. We look forward to incorporating the suggested clarifications in the final version.

---

### Decision · Program_Chairs · 2026-04-30

**Decision:**

Accept (regular)

**Comment:**

This paper addresses an important problem in machine learning interatomic potentials: obtaining reliable uncertainty estimates with minimal additional cost. Reviewers agreed that the proposed BLIP framework is practically relevant and technically well motivated. In particular, the method offers a scalable single-model alternative to ensembles, applies across several message-passing MLIP architectures, and is supported by a strong empirical evaluation spanning training from scratch, fine-tuning, small-data regimes, and active learning.

A major strength of the paper is that BLIP improves both predictive performance and uncertainty quality while incurring only modest overhead. The rebuttal further strengthened the paper by clarifying the novelty beyond standard variational dropout, adding the requested shallow-ensemble comparison, and addressing questions about symmetry preservation, training cost, and applicability.

Some limitations remain. The current efficient implementation does not yet directly support higher-order irreducible-representation-based equivariant models such as MACE or UMA, and the active learning evaluation is more of a proof of concept than a full downstream benchmark across all UQ baselines. However, these concerns do not outweigh the overall strength of the contribution.

Overall, this work has clear practical significance for scientific machine learning and provides an effective solution for uncertainty quantification. I recommend acceptance.